# Local adaptation and archaic introgression shape global diversity at human structural variant loci

**Stephanie M Yan[1], Rachel M Sherman[2], Dylan J Taylor[1], Divya R Nair[1], Andrew N Bortvin[1], Michael C Schatz[1,2], Rajiv C McCoy[1]\***

[1]Department of Biology, Johns Hopkins University, Baltimore, Baltimore, United States; [2]Department of Computer Science, Johns Hopkins University, Baltimore, United States

**Abstract** Large genomic insertions and deletions are a potent source of functional variation, but are challenging to resolve with short-read sequencing, limiting knowledge of the role of such structural variants (SVs) in human evolution. Here, we used a graph-based method to genotype long-read-discovered SVs in short-read data from diverse human genomes. We then applied an admixture-aware method to identify 220 SVs exhibiting extreme patterns of frequency differentiation – a signature of local adaptation. The top two variants traced to the immunoglobulin heavy chain locus, tagging a haplotype that swept to near fixation in certain southeast Asian populations, but is rare in other global populations. Further investigation revealed evidence that the haplotype traces to gene flow from Neanderthals, corroborating the role of immune-related genes as prominent targets of adaptive introgression. Our study demonstrates how recent technical advances can help resolve signatures of key evolutionary events that remained obscured within technically challenging regions of the genome.

**\*For correspondence:**
rajiv.mccoy@jhu.edu

**Competing interest:** The authors declare that no competing interests exist.

## Introduction

Rapid global dispersal and cultural evolution have exposed modern humans to a striking diversity of environments, to which they have developed numerous genetic adaptations (*Fan et al., 2016*). The vast majority of genomic research on human adaptation has focused on single-nucleotide polymorphisms (SNPs) and short insertions and deletions (indels) due to their ease of discovery by high-throughput short-read DNA sequencing (e.g., *The 1000 Genomes Project Consortium, 2015*; *The International HapMap Consortium, 2007*). This focus overlooks the potential impacts of larger insertions, deletions, and inversions – collectively termed structural variants (SVs) – whose abundance and complexity are now being appreciated with the advent of long-read DNA sequencing (*Audano et al., 2019*; *Chaisson et al., 2015*; *Sedlazeck et al., 2018a*). SVs impact more total sequence per genome than SNPs and may severely disrupt genes and regulatory elements to confer large effects on gene expression (*Chiang et al., 2017*). The size, modularity, and functional potential of SVs may offer a rich substrate for natural selection, as supported by several specific examples of adaptive insertions and deletions in diet and pigmentation-related genes (*Hsieh et al., 2019*; *Kothapalli et al., 2016*; *Perry et al., 2007*; *Saitou and Gokcumen, 2019*), as well as genome-wide evidence from primarily short-read data (*Almarri et al., 2020*; *Sudmant et al., 2015*).

More comprehensive analysis of SV evolution has been limited by the long and repetitive nature of many SVs. Short-read approaches to SV discovery depend on abnormalities in depth of coverage or other characteristics of read alignments (*Kosugi et al., 2019*) but achieve sensitivity of only 10–50% (*Chaisson et al., 2019*; *Jakubosky et al., 2020b*). In contrast, the recent development of long-read

sequencing methods has revolutionized SV discovery, achieving high sensitivity and specificity by spanning SV sequences and their unique flanking regions (*Sedlazeck et al., 2018a*). This application of long-read sequencing to human samples has dramatically expanded the catalog of known human variation, generating databases of hundreds of thousands of SVs discovered in globally diverse individuals (*Audano et al., 2019*; *Ebert et al., 2021*). Yet with rare exceptions (*Beyter et al., 2021*), these sequencing methods remain impractical for population-scale applications due to their high costs and low throughput.

We addressed this limitation by applying a hybrid approach to integrate SV discovery from long-read sequencing and SV genotyping in short-read sequenced datasets. Using a graph-based method, we genotyped a catalog of long-read discovered SVs in high-coverage short-read sequencing data from the 1000 Genomes Project (1KGP) (*Byrska-Bishop et al., 2021*). By intersecting these genotype data with existing RNA-seq data from human cell lines, we discovered 1121 significant associations between SVs and the expression of nearby genes, including several components of the human leukocyte antigen (HLA) gene cluster and its interaction partners. We then applied an admixture-aware method to identify 220 SVs that exhibit strong allele frequency differentiation across human populations. Among the top hits, we discovered an extreme signature of local adaptation tagged by separate insertion and deletion polymorphisms at the immunoglobulin heavy chain locus, which encodes key components of the adaptive immune system. Alternative alleles defining this haplotype achieve high frequencies in certain southeast Asian populations, but are rare or absent from other global populations. Searching for signatures of archaic introgression within our set of highly differentiated SVs further revealed evidence that the adaptive haplotype entered the modern human population via ancient hybridization with Neanderthals. Together, our work highlights the role of structurally complex and repetitive regions of the genome as hidden sources of functional diversity and evolutionary innovation in the hominin lineage.

## Results

### Graph genotyping of structural variation

To assess the role of SVs in human adaptive evolution, we sought to combine the accuracy of long-read sequencing with the scale and population diversity of short-read sequencing data. To this end, we reanalyzed long-read sequencing data of 15 individuals from five continents (*Audano et al., 2019*) to discover a set of 107,866 SVs (*Figure 1—figure supplement 1*). The diversity of the long-read sample set (sample ancestries: three African, two American, three East Asian, two European, two South Asian, two hydatidiform moles [likely European]) enables SV discovery across the five continental superpopulations of 1KGP. We found that 89,979 (83.4%) of the long-read-discovered variants, including 30,229 (72.3%) common SVs (allele frequency [AF] ≥ 0.05), are not represented in sets of SVs discovered with short-read sequencing by 1KGP or the Human Genome Diversity Project (HGDP) (*Almarri et al., 2020*; *Sudmant et al., 2015*). Despite the much smaller size of the long-read sample set, we were also able to rediscover 66.0% and 17.7% of common SVs found with short reads in the 1KGP and HGDP datasets, respectively, in agreement with results reported in previous studies (*Zhao et al., 2021*; *Figure 1—figure supplement 2* and *Figure 1—figure supplement 3*; see Materials and methods). We expect that the SVs unique to the short-read datasets reflect both differences in the discovery sample set (i.e., many of the long-read sequenced individuals are in 1KGP, while none are in HGDP) and high rates of false positives in short-read SV detection (*Nattestad et al., 2018*). We additionally discovered 2.3× and 3.5× more insertions than 1KGP and HGDP, respectively, with an improved insertion-to-deletion ratio (1.46 in the long-read dataset; 0.40 in 1KGP; 0.37 in HGDP). The dearth of insertions in the short-read SV sets reflects the challenge of insertion calling with short reads, which have difficulty mapping sequences not present in the reference genome (*Aganezov et al., 2020*), while the excess of insertions among long-read SVs reflects a known variant calling bias caused by missing sequences (e.g., collapsed tandem repeats) in the GRCh38 reference (*Aganezov et al., 2021*).

We then used the graph genotyping software Paragraph (*Chen et al., 2019*) to genotype this set of long-read discovered SVs in 2504 high-coverage (30×) short-read-sequenced samples from 1KGP (*Byrska-Bishop et al., 2021*; *Figure 1A*; see Materials and methods). Results from benchmarking, using ground truth SV data from the Genome in a Bottle Consortium (*Zook et al., 2020*), show that

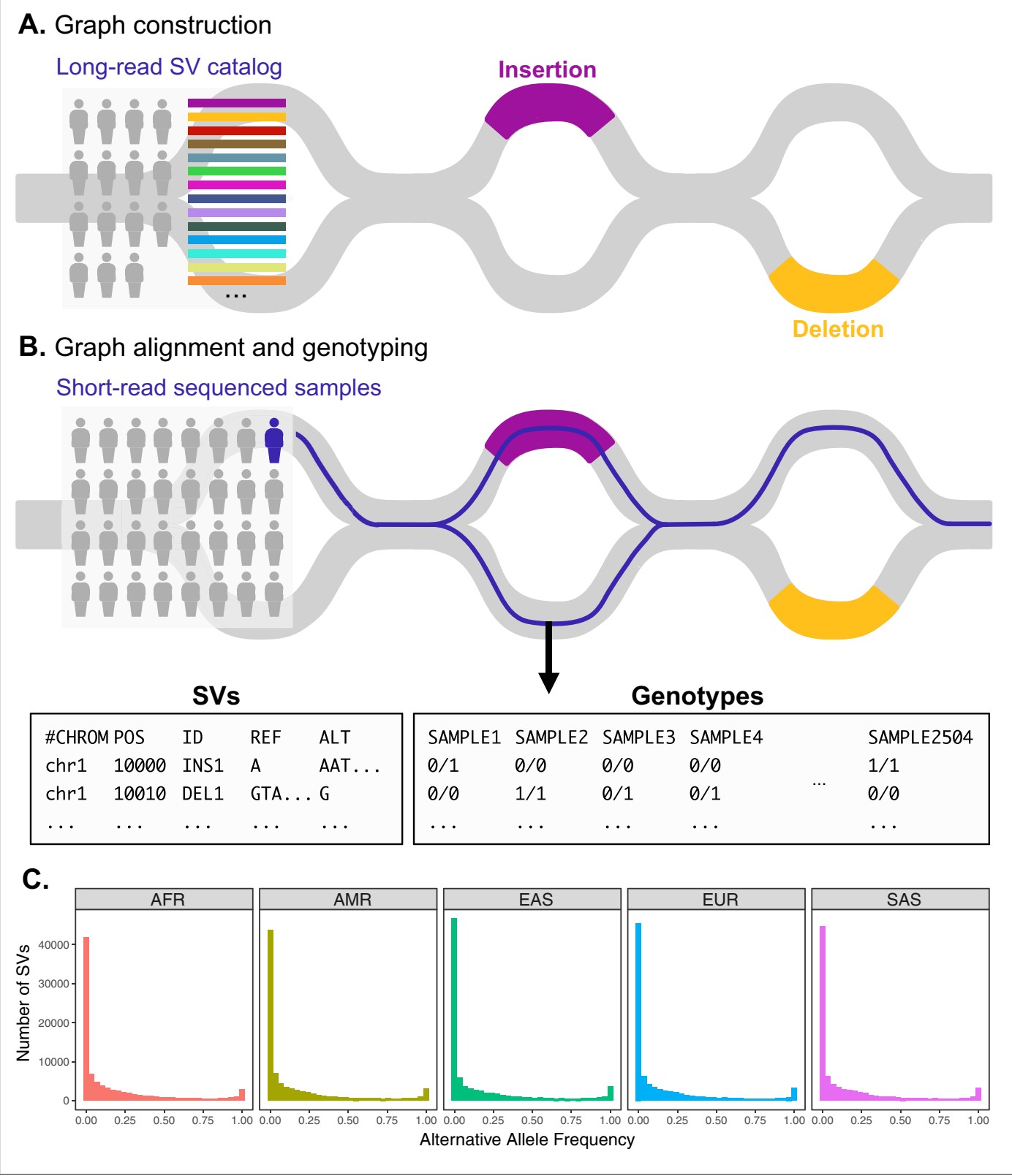

**Figure 1.** Variant graph genotyping of SVs with Paragraph. (**A**) Genotyping of SVs was performed using a graph-based approach that represents the reference and alternative alleles of known SVs as edges. The SVs used for graph construction were originally identified from long-read sequencing of 15 individuals (**Audano et al., 2019**). (**B**) At candidate SV loci, samples sequenced with short reads are aligned to the graph along the path of best fit, and individuals are genotyped as heterozygous (middle), homozygous for the reference allele (right), or homozygous for the alternative allele (not

*Figure 1 continued on next page*

*Figure 1 continued*

depicted). We applied this method to the 1KGP dataset to generate population-scale SV genotypes. (**C**) Allele frequency spectra of SVs genotyped with Paragraph. The left-most bin represents SVs where the alternative allele is absent from the 1KGP sample (AF = 0). Samples are stratified by their 1KGP superpopulation.

The online version of this article includes the following figure supplement(s) for figure 1:

**Source data 1.** Superpopulation-level allele frequencies of structural variants used to construct allele frequency spectra in *Figure 1C*.

**Figure supplement 1.** Distribution of long-read discovered SVs throughout the genome.

**Figure supplement 2.** Allele frequencies (AFs) of SVs discovered in 2504 short-read sequenced samples from 1KGP and in 15 long-read sequenced individuals.

**Figure supplement 3.** Allele frequencies (AFs) of SVs discovered in 929 short-read sequenced samples from the Human Genome Diversity Project (HGDP) and in 15 long-read sequenced individuals.

**Figure supplement 4.** Allele frequencies of SVs genotyped with Paragraph in short-read sequenced samples from 1KGP, compared to the number of PacBio long-read sequenced samples in which the SVs were originally discovered (*Audano et al., 2019*).

**Figure supplement 5.** Global minor allele frequency spectrum, stratified by SV type.

**Figure supplement 6.** Negative relationships between SV length and global minor allele frequency, stratified by SV type.

**Figure supplement 7.** Histogram of LD between SVs and neighboring SNPs and short indels.

among both graph-based and non-graph-based SV genotyping tools, Paragraph consistently attains the best balance of precision and recall for both insertions and deletions (*Chen et al., 2019*; *Hickey et al., 2020*). Paragraph averages precision and recall rates of 0.72 and 0.70, respectively, across all SV types and genomic regions, and an average precision and recall of 0.86 and 0.79 when repeat regions are excluded (*Hickey et al., 2020*), underscoring its effectiveness for SV genotyping. Paragraph achieves such accuracy by generating graph representations of SV loci, which include diverging paths for known alternative alleles such as SVs. Short reads are aligned to the graph along the path of best fit, facilitating genotyping even in structurally complex and repetitive regions (*Figure 1B*). Informed by a large catalog of candidate SV alleles discovered by long-read sequencing, graph genotyping thus permits the study of variants that would be difficult or impossible to discover with short-read data alone (*Chen et al., 2019*; *Hickey et al., 2020*; *Sibbesen et al., 2018*; *Sirén et al., 2021*).

As quality control, we filtered the resulting data based on genotyping rates and adherence to Hardy–Weinberg equilibrium, in accordance with *Chen et al., 2019* (see Materials and methods). Specifically, we removed SVs that were not successfully genotyped in ≥50% of samples, as well as SVs that violated one-sided Hardy–Weinberg equilibrium expectations (excess of heterozygotes) in more than half of the 1KGP populations. The latter scenario, whereby nearly all individuals are genotyped as heterozygous, is a common artifact caused by slightly divergent repetitive sequences that are falsely interpreted as alternative alleles at a single locus (*Graffelman et al., 2017*). These filtering steps removed 15,580 SVs, leaving 92,286 variants for downstream analysis.

Using this remaining set, we examined the allele frequency spectrum both to provide a general description of the data, as well as to quantify the impacts of our unique ascertainment approach (*Figure 1C*, *Figure 1—figure supplement 5*). Global alternative allele frequencies were strongly correlated with the number of long-read-sequenced samples in which they were originally discovered (*Audano et al., 2019*), broadly supporting the accuracy of our graph genotyping results (*Figure 1—figure supplement 4*). A total of 25,201 (27.3%) alternative SV alleles were absent from all 1KGP samples, likely reflecting ultra-rare variation within the panel of long-read sequenced genomes, as well as the imperfect recall (i.e., true positive rate) of SV genotyping (*Chen et al., 2019*). This abundance of rare variation is a known feature of human populations, which have experienced rapid demographic expansion over recent history (*Keinan and Clark, 2012*). In contrast, a total of 1139 (1.2%) alternative SV alleles were observed to be fixed in the 1KGP sample and likely reflect a combination of assembly errors and rare variation in the reference genome. Relaxing the criterion for fixation to 90% alternative allele frequency (thus allowing for a modest rate of false negatives) resulted in a total of 4947 (5.4%) fixed or nearly fixed alternative SV alleles.

The frequency spectrum of segregating variation was meanwhile shaped by an ascertainment bias caused by variant discovery within a small sample and genotyping within a separate and larger sample. This bias, which is similar to that previously described for genotyping microarrays (*Lachance and Tishkoff, 2013*), is characterized by an apparent depletion of rare variation, given that such variants are

not shared with the discovery sample. While important to note, our study should be largely unaffected by this bias, due to our focus on individual variants rather than the shape of the allele frequency spectrum. Moreover, positively selected variants are by definition locally common and thus enriched within a small but globally diverse sample (*Audano et al., 2019*).

We next sought to test whether different classes of SVs exhibited differences in allele frequencies as a potential consequence of purifying selection. Among SVs that were polymorphic within the 1KGP sample, we observed a negative correlation between SV length and minor allele frequency (Kendall's $\tau$ = –0.140, p-value < $1 \times 10^{-10}$), consistent with selection against longer SVs (*Figure 1—figure supplement 6*). We similarly observed that deletions segregated at lower average minor allele frequencies than insertions ($\beta$ = –0.555, p-value < $1 \times 10^{-10}$).

## Quantifying linkage disequilibrium with known SNPs and short indels

One intriguing question regards the extent to which SV genotypes are correlated with those of known SNPs and indels, which has implications for the novelty of any potential discoveries based on the SV data. To address this question, we quantified linkage disequilibrium (LD) between the catalog of long-read discovered SVs and known SNPs and short indels from 1KGP. For each of the 26 1KGP populations, we computed the maximum observed LD between each SV and the nearest 100 variants within a 1 Mb window. Depending on the population subject to measurement, 36–41% of segregating SVs were in strong LD ($r^2 > 0.8$) with any known SNP or short indel, and 52–57% were in moderate LD ($r^2 > 0.5$) with one of these variants (*Figure 1—figure supplement 7*). These observations of maximum LD with SNPs are slightly lower than those computed by *Jakubosky et al., 2020a*, which may reflect our ability to discover SVs in regions that are less accessible to short reads. Levels of LD were lowest for African populations, in accordance with known patterns of haplotype structure (*Conrad et al., 2006*). As with SNPs, these lower levels of LD suggest that the accuracy of SV genotype imputation based on known SNPs will be lowest in African populations, but that resolution for fine-mapping SVs that are potentially causal in phenotypic associations will also be highest (*Wojcik et al., 2019*).

We emphasize that our LD observations are strongly affected by the challenge of small variant discovery and genotyping in repetitive regions of the genome that are enriched for SVs. Specifically, 50,917 of all SVs (47.2%) intersect at least partially with one or more genomic intervals deemed inaccessible based on the 1KGP pilot mask, and 5214 SVs (4.8%) occur within regions of the genome identified as problematic by the ENCODE Consortium (*Amemiya et al., 2019*). Nevertheless, LD with known SNPs and indels is a meaningful metric for our study, because it quantifies the extent to which SVs represent independent and unexplored variation relative to previous evolutionary studies. Low observed levels of LD between a substantial fraction of SVs and known variants presents an opportunity to discover novel functional associations and signatures of adaptation at loci poorly tagged by easily genotyped markers.

## Expression quantitative trait locus (eQTL) mapping

Seeking to first test the functional impacts of common structural variation, we intersected the SV genotype data with RNA-seq data from an overlapping set of 447 samples from the Geuvadis Consortium, which was generated from lymphoblastoid cell lines (LCLs) derived from individuals from four European and one African population (*Lappalainen et al., 2013*). We tested for associations between levels of gene expression and genotypes for SVs within 1 Mb from the transcription start site (TSS). After filtering the data on genotyping call rate, minor allele frequency, and gene expression level (see Materials and methods), we identified a total of 1121 SV-gene pairs with significant gene expression associations at a 10% false discovery rate (FDR; *Figure 2A*), broadly consistent with expectations from previous eQTL studies when scaled to the number of tested SVs (*Chiang et al., 2017*). SVs with significant impacts on expression tended to occur near the genes that they regulate, with 62% of significant SV eQTLs occurring within 100 kb of the corresponding TSS (*Figure 2B*).

Top gene expression associations include several genes in the HLA complex (*HLA-DQB2, HLA-DQA2, HLA-DRB6*, and *HLA-P*), as well as *ERAP2*, which encodes an endoplasmic reticulum aminopeptidase that processes HLA ligands to lengths suitable for their binding (*Figure 2C*). While gene expression data from LCLs provides a limited snapshot of the functional implications of SVs, it is notable that these immune loci also exhibit strong gene expression diversity mediated by genetic variation.

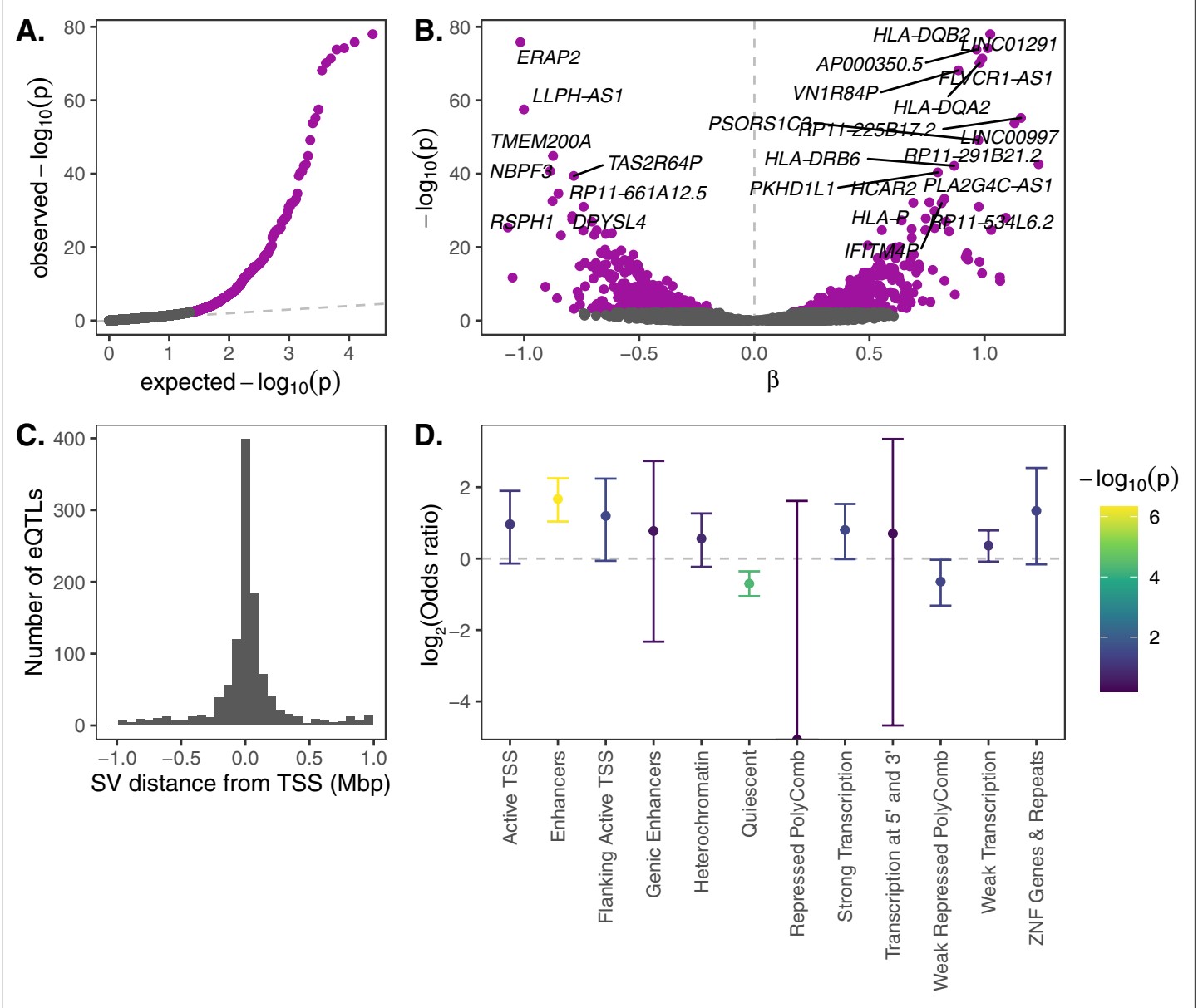

**Figure 2.** eQTL mapping of SVs. We used RNA-seq data from the Geuvadis Consortium (*Lappalainen et al., 2013*), obtained from LCLs derived from individuals from four European and one African population of the 1KGP dataset, to test for associations between SV genotypes and gene expression. SV-eQTL pairs that were significant at a 10 % FDR are depicted in purple. (**A**) Q–Q plot of permutation p-values for all SV-gene pairs tested. (**B**) Volcano plot of eQTLs and the estimated effect of the alternative allele on expression (β). (**C**) Distribution of the distance of significant SV eQTLs from the transcription start site (TSS) of their associated genes. (**D**) Enrichment or depletion of expression associations for SVs that overlap various ChromHMM chromatin state annotations from the Roadmap Epigenomics Project (GM12878 Lymphoblastoid Cells). Chromatin states with ≤0.1 % genome-wide representation were omitted for visual clarity.

The online version of this article includes the following figure supplement(s) for figure 2:

**Source data 1.** Summarized eQTL data underlying *Figure 2*.

**Figure supplement 1.** Relationships between gene expression and genotype for 13 exon-intersecting SV eQTLs.

**Figure supplement 2.** Stacked histogram of the number of variants in each 90 % credible causal set based on fine-mapping of 1121 significant SV eQTLs with CAVIAR (*Hormozdiari et al., 2014*).

We next investigated the mechanisms through which SV eQTLs may impact gene expression. Using epigenetic state annotations generated by the Roadmap Epigenetics Consortium (*Roadmap Epigenomics Consortium et al., 2015*), we found a strong enrichment of significant gene expression associations for SVs that intersect annotated enhancer elements, as well as a strong depletion of expression associations for SVs that intersect 'quiescent' sequences that are devoid of important epigenetic marks (*Figure 2D*). We also further investigated the 13 significant SV eQTLs that intersected one or more exons of their associated gene. Of the nine such deletions, 8 exhibited associations consistent with direct dosage effects on expression (i.e., a linear association between expression and deletion copy number; *Figure 2—figure supplement 1A*). Meanwhile, the three exon-intersecting insertions exhibited associations in both directions (one positive and two negative associations; *Figure 2—figure supplement 1B*). Together, these intuitive results shed light on functional mechanisms through which SVs may mediate impacts on phenotypes and fitness.

Lastly, we performed fine-mapping of SV and SNP eQTLs with CAVIAR (*Hormozdiari et al., 2014*) with the goal of homing in on candidate causal variants at SV eQTL loci. Of the 1121 loci with a significant SV eQTL, 345 (30.8%) contained the SV within the 90% credible causal set. While the median size of the credible causal sets was 19 variants, we identified 15 loci where an SV occurred among a credible causal set of five or fewer variants (*Figure 2—figure supplement 2*). These include a 37 Kb deletion (27005_CHM13_del) that is among just two variants in a credible causal set for expression effects on *MIF-AS1*, a lncRNA partially encompassed by the deletion. The expression of this lncRNA has been previously associated with breast and gastric cancer proliferation (*Ding et al., 2019*; *Li et al., 2018*). Similarly, a 3 kb insertion (20290_HX1_ins) is among two variants in a credible causal set for expression effects on *CSF2RB*, the dysfunction of which has been associated with pulmonary disease (*Suzuki et al., 2011*). While this approach offers a roadmap for future fine-mapping studies investigating the relative impacts of different variant classes on gene expression, we caution that the present analysis may still underestimate the potential causal role of SVs, due to a modestly higher error rate for genotyping of SVs versus SNPs (*Chen et al., 2019*).

## Admixture-aware scan for signatures of local adaptation

We next sought to use our set of population-scale SV genotypes to search for SVs with signatures of local adaptation. Such locus-specific scans for local adaptation can be broadly classified into frequency differentiation-based and LD-based approaches (*Vitti et al., 2013*). While powerful, LD-based methods generally require haplotype phasing and/or dense and accurate genotyping – a requirement that is infeasible for many of the complex and repetitive regions of the genome investigated in our study. In contrast, frequency differentiation-based approaches can be applied to individual loci and are based on the logic that positive selection tends to cause a particular allele to increase in frequency only in the population(s) where it became established and was advantageous. However, a limitation of these methods is the requirement that individuals are grouped into pre-defined populations that may not reflect genetic patterns or the substantial admixture exhibited by many human populations.

To overcome these limitations, we used Ohana (*Cheng et al., 2017*; *Ilardo et al., 2018*; *Cheng et al., 2019*), a maximum likelihood-based method that models individuals as possessing ancestry from combinations of *k* ancestry components, inspired by related methods (*Alexander et al., 2009*; *Falush et al., 2003*). The method then tests whether individual variants adhere to this genome-wide null model, or are better explained by an alternative model in which frequencies are allowed to vary in one or more populations by consequence of local adaptation. Following the precedent set by 1KGP (*The 1000 Genomes Project Consortium, 2015*), we modeled individual genomes as combinations of eight ancestry components, replicating known patterns of population structure at continental scales as well as prominent signatures of admixture within a subset of populations (e.g., African ancestry in Southwest US [ASW] and admixed American populations [AMR]; *Figure 3A*). The distribution of ancestry components across populations was qualitatively unaffected by the choice of *k* (*Figure 3—figure supplement 1*).

Across all ancestry components, we identified 220 unique SVs exhibiting significant deviation from genome-wide patterns of allele frequency differentiation (99.9th percentile of matched distribution for SNPs and short indels; see Materials and methods; *Figure 3B*; *Supplementary file 1*; *Figure 3—figure supplement 1*). These included 139 SVs with coordinates overlapping with annotated genes, of which 13 intersected with annotated exons. Seven SVs at these frequency-differentiated loci were also

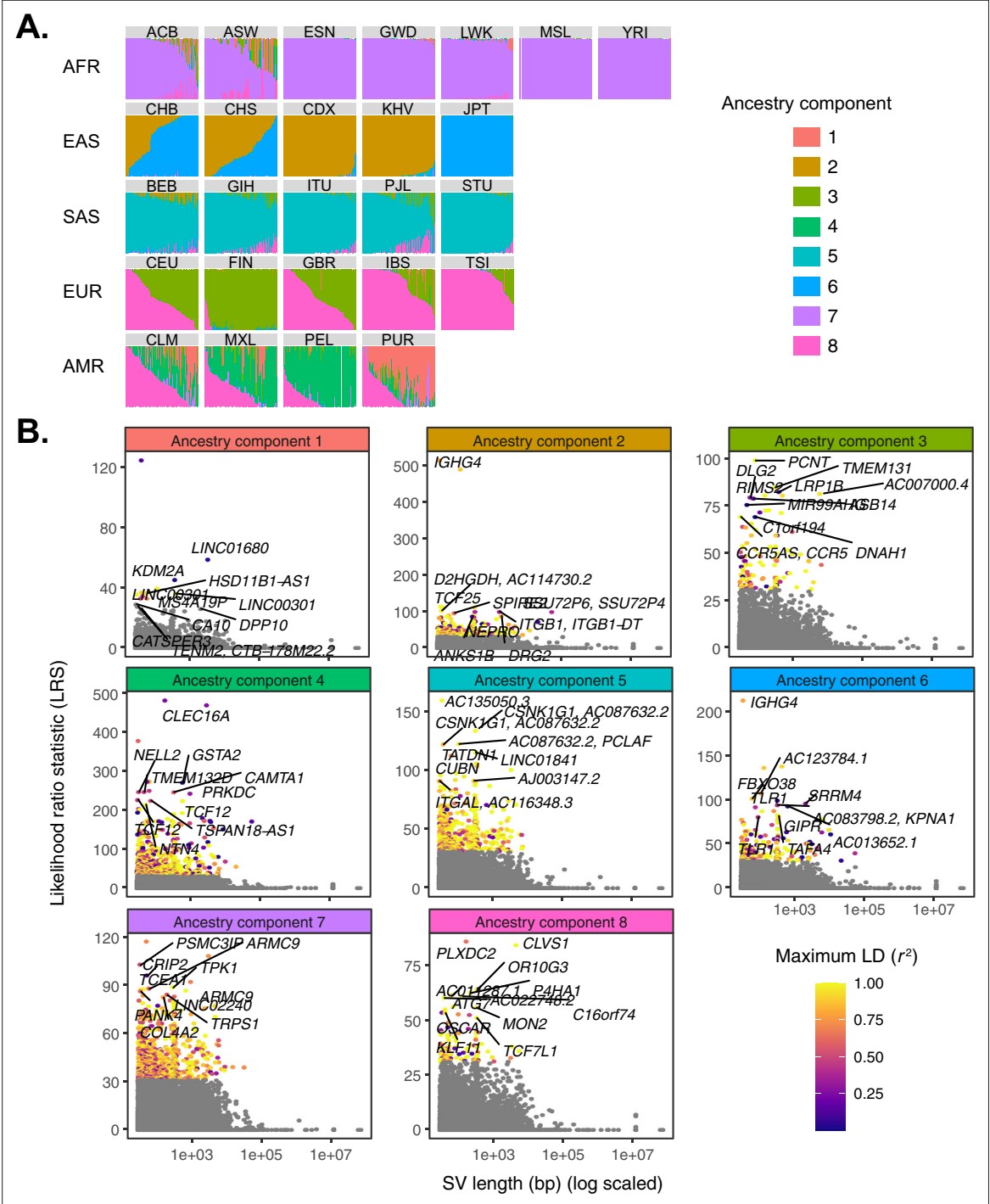

**Figure 3.** Quantifying allele frequency differentiation among ancestry components. (**A**) To conduct an admixture-aware scan for local adaptation, we used Ohana (*Cheng et al., 2017*; *Ilardo et al., 2018*) to infer genome-wide patterns of ancestry in the 1KGP samples. This method models each individual as a combination of *k* ancestry components and then searches for evidence of local adaptation on these component lineages. Admixture proportions (*k* = 8) for all samples in the 1KGP dataset, grouped by population. Vertical bars represent individual genomes. (**B**) Using Ohana, we searched for evidence of local adaptation by testing whether the allele frequencies of individual variants were better explained by the genome-wide covariance matrix, or by an alternative covariance matrix where allele frequencies were allowed to vary in one ancestry component. The likelihood ratio statistic (LRS) reflects the relative support for the latter selection hypothesis. For each ancestry component, SVs with LRS > 32 are colored by their maximum linkage disequilibrium ($r^2$) with any known SNP in the corresponding 1KGP superpopulation.

The online version of this article includes the following figure supplement(s) for figure 3:

**Source data 1.** Ancestry component results underlying *Figure 3A*.

*Figure 3 continued on next page*

**Figure supplement 1.** Comparison of admixture proportions inferred by Ohana for samples in the 1KGP dataset.

**Figure supplement 2.** Visualization of the likelihood ratio statistic for SNPs and short indels in the genomic vicinity of the top SV hit for each ancestry component.

significant eQTLs based on our previous analysis of the Geuvadis LCL data (*Supplementary file 2*). Through comparison to results for SNPs and indels, we found that 88 of the outlier SVs we identified were among the top 10 most frequency-differentiated variants within a 1 Mb window, suggesting that these represent examples where SVs may be causal targets of selection at adaptive loci. Finally, only 119 (54.1%) highly differentiated SVs possessed strong LD with known SNPs or short indels ($r^2 > 0.8$; *Figure 1—figure supplement 2*), indicating that many of these loci constitute novel candidate targets of selection that may have been missed by previous scans.

## Known and novel targets of local adaptation

Notable examples of highly differentiated loci included rs333, the Δ32 allele of the chemokine receptor *CCR5*, which is known to confer resistance to HIV infection and progression (*Dean et al., 1996*). Among our results, this deletion polymorphism is the 14th most frequency-differentiated SV with respect to ancestry component 3, which is highly represented in Europe. The *CCR5-Δ32* allele segregates at moderate frequencies in European populations (MAF = 10.9%) and achieves its highest frequency of 15.6% in the Finnish population, but segregates at low frequencies elsewhere. The case for historical positive selection at this locus has been contentious, with initial studies citing a geographic cline in allele frequencies and strong LD with adjacent microsatellite markers (*Stephens et al., 1998*), potentially driven by epidemics such as the bubonic plague or smallpox (*Galvani and Slatkin, 2003*). However, subsequent studies argued that patterns of long-range LD (*Sabeti et al., 2005*) and temporal allele frequency changes based on ancient DNA (aDNA) samples (*Bollback et al., 2008*) could not exclude models of neutral evolution.

Our study also identified numerous novel hits, such as a 309 bp intronic insertion in *TMEM131* (ancestry component 3), which segregates at a frequency of 63% in Finnish populations, but at a mean frequency of 24% in non-European populations. This gene encodes proteins involved in collagen cargo trafficking from the endoplasmic reticulum to the Golgi (*Zhang et al., 2020*). Collagen is the most abundant protein in the human body and the major component of human skin. Recurrent positive selection shaping skin pigmentation and other phenotypes is one of the best described examples of local adaptation across human populations (*Crawford et al., 2017*; *Jablonski, 2004*).

We also identified a 2.8 kb insertion in an intron of *CLEC16A*, inferred to be under selection in ancestry component 4, which corresponds to the Peruvian (PEL), Mexican (MXL), and Colombian (CLM) populations of 1KGP. The insertion, which is among the strongest frequency-differentiated variants at its respective locus and segregates in LD with several SNPs and short indels (maximum $r^2$ = 0.81; *Figure 3—figure supplement 3*), occurs at frequencies of 52.4%, 38.3%, and 15.4%, respectively, in these three populations, but is rare in others (AF < 0.04). *CLEC16A* is thought to impact susceptibility to autoimmune disorders, and SNPs in this gene have been associated with diseases such as type I diabetes, multiple sclerosis, and rheumatoid arthritis (*Pandey et al., 2019*). Notably, this same SV was also identified in a recent study of structural variation, which similarly reported that it achieves high frequency in Peruvian populations (*Ebert et al., 2021*).

In rare cases, multiple SVs in LD with one another captured the same underlying signature of frequency differentiation. One such example from South Asian populations (ancestry component 5) involved a linked intronic insertion and deletion in the cellular growth and morphogenesis related gene *CSNK1G1*. In this case, the reference genome carries the global minor allele, such that the signature of local adaptation presents as a lower frequency of the alternative allele in South Asian populations (60–71% for 22980_HG00514_ins; 60%–72% for 25014_HG02106_del) compared to other global populations (91% and 92%, respectively).

## Extreme signatures of adaptation at the immunoglobulin locus in southeast Asian populations

The SV with the strongest evidence of local adaptation across all populations was an insertion polymorphism in an intron of *IGHG4*, which codes for a constant domain of the immunoglobulin heavy chain. These heavy chains pair with light chains, the latter of which include a domain composed of variable (V), diversity (D), and joining (J) segments. Complementing their substantial germline variation, V(D)J loci experience somatic recombination and hypermutation to generate vast antibody repertoires – the defining feature of the adaptive immune system (*Watson et al., 2017*). This insertion polymorphism identified by our scan exhibits strong allele frequency differentiation in ancestry component 2, which is highly represented in the Chinese Dai in Xishuangbanna, China (CDX) and Kinh in Ho Chi Minh City, Vietnam (KHV) populations, where it achieves frequencies of 88% and 65%, respectively, while remaining at much lower allele frequencies in other global populations (*Figure 4A,B*).

The SV was originally reported as a 34 bp insertion based on long-read sequencing of the genome of a Vietnamese individual (HG02059) (*Audano et al., 2019*). Based on realignment to a modified version of the reference genome that includes the alternative allele, we revised the sequence of this insertion to 33 bp, but found that it is well supported by patterns of coverage, split reads, and soft clipped alignments at the SV breakpoints (*Figure 4—figure supplement 1*). The sequence of the insertion itself is repetitive within the human reference genome, with two identical copies of the sequence occurring ~44 and ~117 kb downstream, respectively, also within the *IGH* gene cluster. This SV is included among sets of indels called by gnomAD (*Karczewski et al., 2020*), 1KGP, and HGDP, with a slightly altered position and insertion sequence. However, it exhibits a highly skewed allele balance (centered on 0.2, in contrast to the expectation of 0.5) – a symptom of reference bias in read mapping (*Chen et al., 2019*) – as well as low genotyping rates (31% for CDX in 1KGP, compared with 85% in our study). These biases result in lower estimates of the insertion's allele frequency in East Asian populations (AF = 0.73 in CDX) and underscore the technical challenges of genotyping at this locus with traditional short-read approaches (*Chin et al., 2020*; *Zhang et al., 2021*).

Interestingly, the second strongest signature of adaptation across all populations traced to a nearby 135 bp deletion, which overlaps with two transcription factor binding sites for *IGHE*, another component of the constant region of immunoglobulin heavy chains. This deletion, which lies only 33 kb upstream of the *IGHG4* insertion (*Figure 4A*), again achieves high frequencies in the CDX and KHV populations (70% and 54%, respectively) but segregates at low frequencies in most other global populations. Despite their genomic proximity and similar patterns of frequency differentiation, these two SVs exhibit only modest levels of LD ($r^2 = 0.17$), likely reflecting recombination occurring during or after the episode of selection.

## Evidence of Neanderthal-introgressed origin of the high-frequency IGH haplotype

Approximately 2% of the genome of all non-African individuals traces to admixture with Neanderthals between 47 and 65 kya, while Oceanian populations (and Asian populations to a lesser extent) also possess sequences introgressed from Denisovans (*Green et al., 2010*; *Sankararaman et al., 2012*; *Sankararaman et al., 2016*; *Vernot et al., 2016*). Introgressed alleles, including some SVs (*Almarri et al., 2020*; *Hsieh et al., 2019*), are thought to have conferred both beneficial and deleterious effects on modern human populations, especially with respect to immune-related phenotypes (*Rotival and Quintana-Murci, 2020*). Motivated by these findings, we tested our set of highly differentiated SVs for evidence of archaic hominin introgression. Using results from the method Sprime (*Browning et al., 2018*), which leverages patterns of divergence and haplotype structure to classify archaic introgressed sequences, we identified SVs that segregate in LD with putative introgressed SNPs (see Materials and methods). Of the 220 highly differentiated SVs, 26 (12%) exhibited moderate LD ($r^2 > 0.5$) with putative introgressed haplotypes, while simultaneously exhibiting low allele frequencies (AF < 0.01) within African populations of 1KGP (*Figure 4—figure supplement 2*; *Supplementary file 3*). Notably, this set of candidate introgressed SVs included the *IGHG4* insertion and nearby deletion, which, despite their low LD with one another, each tag multiple putative introgressed SNPs within the CDX population. Indeed, the original Sprime publication reported that putative introgressed variants of *IGHA1*, *IGHG1*, and *IGHG3* achieve high frequencies in Eurasian populations (*Browning et al., 2018*). The variants identified by *Browning et al., 2018* are located in a subregion of the broader *IGH* locus,

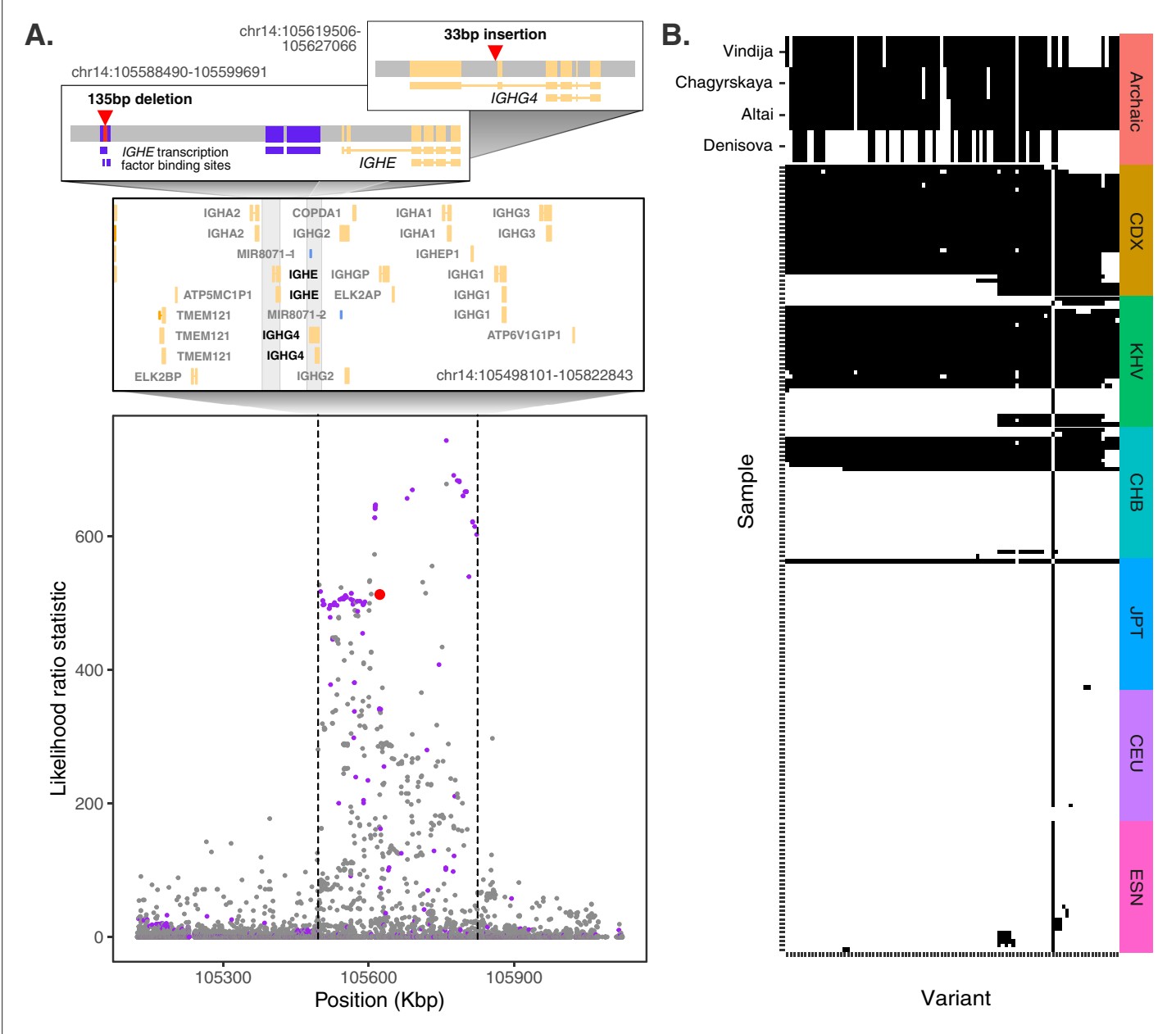

**Figure 4.** Evidence for Neanderthal introgression of the adaptive *IGH* haplotype. (**A**) Local analysis of likelihood ratio statistics (LRS) in the region near the 33 bp insertion (red point) reveals a 325 kb haplotype encompassing 94 SNPs with strong allele frequency differentiation within ancestry component 2. Points where the alternative allele matches an allele observed in the Chagyrskaya Neanderthal genome but at a frequency of 1 % or less in African populations are highlighted in purple. (**B**) Individual haplotypes defined by the highly differentiated SNPs (LRS > 450). Four archaic hominin genomes are plotted at the top, while 30 randomly sampled haplotypes from each of 6 populations from 1KGP are plotted below. Archaic hominins samples are colored according to whether they possess more than one aligned read supporting the alternative allele at a given site. ESN refers to the Esan in Nigeria population of 1KGP. Other 1KGP population codes are provided in the main text.

The online version of this article includes the following figure supplement(s) for figure 4:

**Figure supplement 1.** Visualization of read alignments to a modified version of the reference genome at the *IGHG4* locus including the intronic 33 bp insertion sequence.

**Figure supplement 2.** Distribution of LD between adaptive SVs and introgressed SNPs called by Sprime.

**Figure supplement 3.** Population-specific allele frequencies in the broader *IGH* region.

**Figure supplement 4.** Filtering of Neanderthal-introgressed alleles at the *IGH* locus.

**Figure supplement 5.** Population distribution of *IGHG4* insertion using a diagnostic sequence.

*Figure 4 continued on next page*

*Figure 4 continued*

**Figure supplement 6.** Signatures of archaic introgression based on calls from Sprime.

**Figure supplement 7.** Local LD at the *IGH* locus.

**Figure supplement 8.** Global allele frequencies of the introgressed *IGH* haplotype.

---

downstream of the introgressed SVs, and segregate at high allele frequency in East Asian, European, and American populations (*Figure 4—figure supplement 3*). The southeast Asian-specific haplotype we identify, which includes the *IGHG4* insertion and nearby deletion, may have been challenging to discover due to the difficulties of short-read alignment and genotyping in this region of the genome (*Zhang et al., 2021*). Indeed, 80.7% of the sequence in the broader *IGH* locus was filtered out by *Browning et al., 2018* through strict masking of aDNA genotypes to remove low-coverage, poorly mapping, or repeat-associated reads (*Figure 4—figure supplement 4*).

Most candidate introgressed SVs fall within complex regions of the genome, such that genotyping directly from fragmented and degraded aDNA data remains infeasible. However, the *IGHG4* intronic insertion is short enough to be spanned by individual sequencing reads, thus allowing us to validate our genotyping results and further investigate introgression at this locus by directly searching for an exact match to a diagnostic sequence. Specifically, we identified a 48 bp sequence that is unique to individuals with the insertion, but not observed in the reference genome (see Materials and methods), thereby facilitating rapid searches for the insertion allele in any sequencing dataset. Application of this approach to the 1KGP data broadly validated our graph genotyping results and supported the strong allele frequency differences that we had previously described (*Figure 4—figure supplement 5*). We then extended this scan to published short-read sequencing data from four high-coverage (~30×) archaic hominin genomes (*Mafessoni et al., 2020*; *Meyer et al., 2012*; *Prüfer et al., 2017*; *Prüfer et al., 2014*). While the diagnostic sequence was absent from the Denisovan genome, we observed one or more perfect matches in the sequences of each of three Neanderthals, which we found notable given its absence among African populations (*Figure 4—figure supplement 5*).

We next expanded our analysis to the genomic region around the *IGHG4* insertion, investigating the archaic hominin allelic states at each of the 109 highly differentiated variants (2 SVs, 14 short indels, and 93 SNPs) defining the 325 kb LRS peak at the *IGH* locus. Focusing on the 93 SNPs where archaic alleles are easily compared, we observed that the Denisovan genome exhibited a modest degree of matching (39 shared alleles [42%]), while the Altai, Vindija, and Chagyrskaya Neanderthal genomes exhibited near perfect matching over the entirety of the differentiated region (77 [83%], 88 [95%], and 89 [96%] shared alleles, respectively) (*Figure 4B*). Additionally, of these 93 highly differentiated SNPs, 64 were called as introgressed in CDX by Sprime (*Figure 4—figure supplement 6*). Combined with the near absence of these alleles from other global populations and the length of the differentiated haplotype, our observations strongly support the conclusion that this sequence originated via ancient introgression from a Neanderthal population related to the Chagyrskaya and Vindija Neanderthals.

Examination of the haplotype structure at the *IGH* locus revealed four discrete blocks of LD within the highly differentiated region (*Figure 4—figure supplement 7*), consistent with substantial recombination after the original haplotype achieved high frequency. The deletion SV (22231_HG02059_del) falls within the largest LD block, while the insertion SV (22237_HG02059_ins) falls within a smaller LD block that exhibits the greatest allele frequency differences. The latter block includes the tag SNP rs150526114, where the global minor allele matches the Neanderthal genomes and segregates at 91% frequency in CDX, 73% frequency in KHV, and 59% frequency in CHS, but is rare or absent in most other populations from 1KGP. Data from HGDP (*Bergström et al., 2020*) and the Simons Genome Diversity Panel (SGDP) (*Mallick et al., 2016*) shed additional light on the geographic distribution of the putative Neanderthal-introgressed allele and confirmed its extreme pattern of frequency differentiation specific to southeast Asian populations (*Figure 4—figure supplement 8*). Notably, the allele is absent from the HGDP populations from the Americas, which are thought to have split from East Asian populations approximately 26 kya (*Moreno-Mayar et al., 2018*), further supporting the recency and geographically restricted nature of this positive selection event.

## Strength and timing of adaptive introgression at the IGH locus

The challenge of dense genotyping and phasing within the broader *IGH* region (*Zhang et al., 2021*) hinders haplotype-based approaches for inferring the timing of selection. Nevertheless, the global patterns of allele frequencies we observe (*Figure 5A,B*) can be interpreted in the context of known population demographic histories, providing a rough estimate of such timing. For example, pairwise divergence times among East Asian populations inferred by *Wang et al., 2018* constrain the plausible timing of selection at the *IGH* locus between approximately 60 generations (1740 years) and 400 generations (11,600 years) ago (assuming a 29 -year generation time *Tremblay and Vézina, 2000*): after the divergence of the Chinese Dai and Vietnamese populations from the Japanese (JPT) population, but before the divergence of CDX and KHV. These divergence time estimates are roughly consistent with those inferred using an alternative approach based on the joint allele frequency spectrum by *Jouganous et al., 2017*.

To achieve further insight into this episode of selection, we conducted forward genetic simulations of a simplified five-population demographic model based on parameters obtained from the literature (*The 1000 Genomes Project Consortium, 2015*; *Jouganous et al., 2017*; *Wang et al., 2018*; *Figure 5C*). For simplicity, we began the simulations 46 kya, after Neanderthal introgression has occurred and the introgressed haplotype is segregating within the modern human population. Our model makes an additional simplification by not including migration, but we note that this omission should make our estimates of the selection coefficient conservative, as stronger selection is required to generate allele frequency differences between populations exchanging migrants. We first performed 100,000 neutral simulations (selection coefficient [$s$] = 0), finding that the magnitudes of allele frequency differentiation, especially in southeast Asia, far exceed those expected under the null (*Figure 5—figure supplement 1*). In order to formally estimate the parameters of the selection event, we next applied approximate Bayesian computation (ABC) to an additional 175,000 simulations in which we varied relevant parameters (see Materials and methods). In line with our previous intuition, the results of this analysis indicated a recent onset of extremely strong selection, with the selection coefficient parameter ($s$) inferred to be 0.06 (95% credible interval [CI] [0.02, 0.12]), the selection onset time parameter ($T_{adaptive}$) inferred to be 4400 years ago (95% CI [1700, 8400]), and the initial frequency of the adaptive allele in the ancestral Eurasian population ($p_0$) to be 0.18 (95% CI [0.04, 0.35]) (*Figure 5D–F*). We observed a strong correlation between estimates of the selection coefficient and timing of selection. Specifically, older, weaker selection could produce the same frequency differences as stronger and more recent selection (*Figure 5G*), though even the lower bound on our estimate of $s$ places it among the strongest episodes of positive selection documented in humans.

## Discussion

Long-read sequencing is starting to provide more comprehensive views of the landscape of human genetic variation, drawing novel links to phenotypes and diseases. However, long-read sequencing methods remain impractical for most population-scale studies due to their low throughput and high cost. We sought to overcome these limitations by applying variant graph genotyping of a large catalog of long-read-discovered SVs to short-read sequencing data from globally diverse individuals of 1KGP. By mapping eQTLs and scanning for evidence of local adaptation and adaptive introgression, we highlighted the role of SVs as largely unexplored contributors to variation in human genome function and fitness.

Despite the scale and diversity of SVs and samples used in our study, we anticipate that additional SV targets of selection remain undiscovered, either because they were not present in the set of long-read sequenced individuals or because they remain inaccessible to genotyping using graph-based approaches (e.g., tandem repeats) (*Chen et al., 2019*). Studying the evolutionary impacts of such SVs will require the application of long-read sequencing at population scales (*Ebert et al., 2021*), which is still infeasible for most genetics studies. Moreover, SV loci exhibiting expression associations or signatures of selection may not themselves be the causal targets, but rather tag nearby causal variation by consequence of LD. Methods such as fine mapping (*Schaid et al., 2018*) and multiplex reporter assays (*Tewhey et al., 2018*; *van Arensbergen et al., 2019*) will be invaluable for disentangling LD to reveal causal relationships and contrast the relative impacts of various forms of genetic variation.

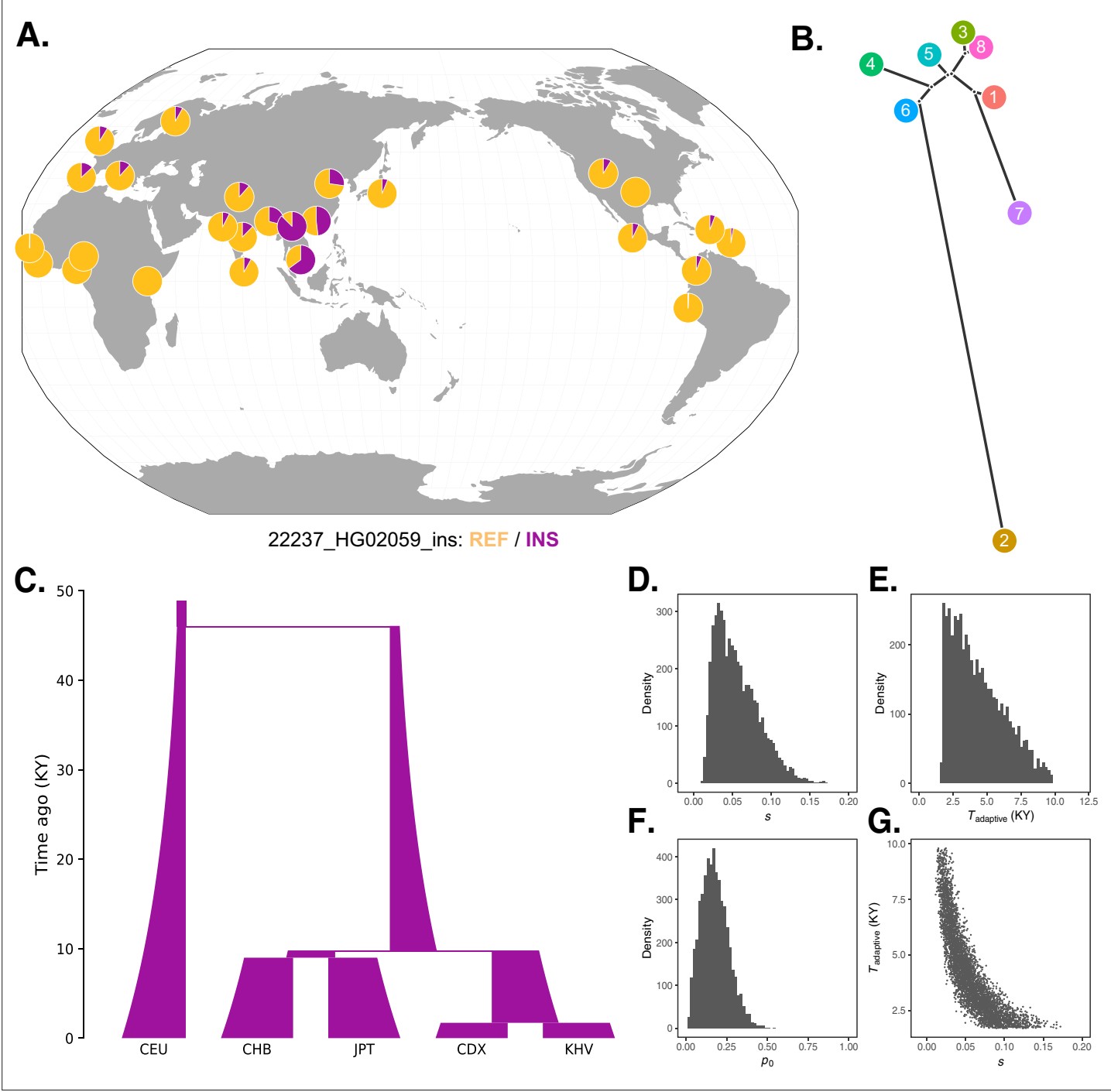

**Figure 5.** Local adaptation at the *IGH* locus. (**A**) Population-specific frequencies of the putative Neanderthal-introgressed insertion allele in each of the 1KGP populations, in the style of the Geography of Genetic Variants browser (*Marcus and Novembre, 2017*). (**B**) Tree representation of the best-fit selection hypothesis for Neanderthal-introgressed haplotype tagged by the *IGHG4* insertion polymorphism, as computed by Ohana. (**C**) Five-population demographic model used for simulation and parameter inference via approximate Bayesian computation (ABC). Population sizes and split times are further described in Materials and methods. (**D**) Posterior distribution of the selection coefficient ($s$). (**E**) Posterior distribution of the timing of the onset of selection ($T_{adaptive}$). (**F**) Posterior distribution of the initial allele frequency at the beginning of the simulation ($p_0$). (**G**) Negative relationship between $s$ and $T_{adaptive}$ for simulations retained by ABC.

The online version of this article includes the following figure supplement(s) for figure 5:

**Source data 1.** Parameters and summary statistics for the simulations retained in the ABC analysis, underlying *Figure 5D–G*.

**Figure supplement 1.** Distribution of $F_{ST}$ and population branch statistic (PBS) for the *IGHG4* insertion from 100,000 neutral simulations.

*Figure 5 continued on next page*

*Figure 5 continued*

**Figure supplement 2.** Annotated Q–Q plot from a phenome-wide association analysis of putative Neanderthal-introgressed variants at the *IGH* locus, using summary statistics from East Asian individuals from the pan-ancestry GWAS of the UK Biobank dataset.

The two strongest signatures of local adaptation in our study traced to the immunoglobulin heavy chain locus. While the precise phenotypic impacts of these variants remain unknown (*Figure 5—figure supplement 2*), their potential effects on adaptive immunity is intriguing given the established role of immune-related genes as common targets of local adaptation in human populations (*Barreiro and Quintana-Murci, 2020*). The human *IGH* locus is highly polymorphic (*Mikocziova et al., 2020*), with examples of SNPs and copy number variants exhibiting frequency differences between populations (*Watson et al., 2013*). In developing lymphocytes, this locus undergoes somatic V(D)J recombination and hypermutation to produce antibodies that drive the immune response—processes that may be influenced by nearby germline variation (*Watson et al., 2017*). The combination of these forms of variation makes the region difficult to probe with traditional sequencing methods (*Chin et al., 2020*; *Zhang et al., 2021*), in turn highlighting the power of long-read sequencing and graph genotyping.

Our observation of the *IGHG4* insertion within the Neanderthal genomes allowed us to connect and build upon initial evidence of selection and introgression at the *IGH* locus. Specifically, 1KGP previously reported SNPs in several immunoglobulin genes as allele frequency outliers with respect to the CDX population (*The 1000 Genomes Project Consortium, 2015*). *Browning et al., 2018* later noted that a Neanderthal-introgressed haplotype at the *IGH* locus achieves high frequencies in Eurasian populations, although they focused on a separate haplotype in the region, split by recombination, that is not specific to southeast Asia. Our findings further expand the evolutionary history of this region of the genome, as well as adding to a growing list of examples of adaptive introgression from archaic hominins (*Gittelman et al., 2016*; *Hsieh et al., 2019*; *Huerta-Sánchez et al., 2014*; *Racimo et al., 2017*), several of which are thought to have targeted immune-related phenotypes (*Abi-Rached et al., 2011*; *Dannemann et al., 2016*; *Enard and Petrov, 2018*; *Gouy and Excoffier, 2020*; *Mendez et al., 2012a*; *Mendez et al., 2012b*; *Sams et al., 2021*).

Based on forward genetic simulations, we estimated that selection on the introgressed *IGH* haplotype initiated between 1700 and 8400 years ago, before the divergence of the CDX and KHV populations and in line with our intuition based on patterns of allele frequencies. This recent onset of selection is intriguing given that introgression from Neanderthals into the ancestors of Eurasian modern human populations dates to 47–65 kya (*Sankararaman et al., 2012*). Our findings thus suggest that persisting archaic introgressed haplotypes provided a reservoir of functional variation to the ancestors of CDX and KHV that proved adaptive during a period of environmental change, for example in response to local pathogens (*Rasmussen et al., 2015*). Recent reports that Neanderthal-introgressed sequences mediate individual outcomes of SARS-CoV-2 infections to this day lend plausibility to this hypothesis (*Zeberg and Pääbo, 2021*; *Zeberg and Pääbo, 2020*; *Zhou et al., 2021*), as does polygenic evidence of adaptation in response to ancient viral epidemics, including in East Asia (*Souilmi et al., 2021*). Moreover, the delayed onset of selection inferred in our study is consistent with recent evidence of time lags between introgression and adaptation favoring other Neanderthal-introgressed alleles (*Yair et al., 2021*).

Our simulations additionally demonstrated that a selection coefficient between 0.02 and 0.12 best explains the observed frequency differences, comparable to other episodes of strong selection in humans. These include examples such as lactase persistence mutations near *LCT* (0.01–0.15) (*Bersaglieri et al., 2004*) and malaria resistance mutations affecting *DARC* (0.08) (*Hamid et al., 2021*) and *HBB* (0.1) (*Elguero et al., 2015*). We caution against overinterpretation of these parameter estimates given the uncertainty in the underlying demographic model, which relies on estimates of population size and divergence time from two studies (*Jouganous et al., 2017*; *Wang et al., 2018*) and does not incorporate admixture between populations. However, our results are broadly consistent with the observation of extreme allele frequency differences among closely related populations in East Asia. Future studies incorporating more complex evolutionary models and fully resolved *IGH* haplotypes (*Rodriguez et al., 2020*) will be essential for further refining the complex evolutionary history of selection and recombination at the immunoglobulin locus. Nevertheless, the coexistence of V(D)J recombination, somatic hypermutation, and local adaptation at this locus presents a remarkable

example of diversifying selection at multiple scales of biological organization, generating allelic diversity both within individuals and across populations.

Together, our study demonstrates how new sequencing technologies and bioinformatic algorithms are facilitating understanding of complex and repetitive regions of the genome – a new frontier for human population genetics. Our study also demonstrates how the integration of short and long-read datasets allows for investigation of these regions at population scales. Combined with studies of diverse populations, these technologies are providing a more complete picture of human genomes and the evolutionary forces by which they are shaped.

# Materials and methods

## Structural variant calling and comparison with short-read datasets

We used published long-read sequencing data from 15 individuals to generate a set of 107,866 SVs for graph genotyping (*Audano et al., 2019*). Raw reads were downloaded using the accessions provided in the original publication and aligned with NGM-LR (*Sedlazeck et al., 2018b*) using default PacBio parameters to the main chromosomes of GRCh38. Variants were called with Sniffles (*Sedlazeck et al., 2018b*), requiring a minimum SV length of 30 bp and a minimum of 10 supporting reads. Resulting VCFs were refined with Iris (*Alonge et al., 2020*) to polish the reported SV sequences. Variants were then merged with SURVIVOR v1.0.7 (*Jeffares et al., 2017*), using a merge distance of 50 bp and requiring strand and type to match. For each merged variant, a representative variant was then obtained from the original pre-merged call set to improve accuracy. Such representative variants were selected by first prioritizing homozygous over heterozygous calls, and then by prioritizing variants with greater proportions of reads supporting the non-reference allele. To prepare the variants for input into Paragraph, translocations, mitochondrial DNA variants, inversions and duplications over 5 kb, and variants without a 'PASS' filter were removed from the VCF. This resulted in a set of 107,866 SVs.

We compared this set of long-read discovered SVs with SVs discovered from short-read sequencing in 1KGP (*Sudmant et al., 2015*; http://ftp.1000genomes.ebi.ac.uk/vol1/ftp/phase3/integrated_sv_map/supporting/GRCh38_positions/) and HGDP (*Almarri et al., 2020*; ftp://ngs.sanger.ac.uk/production/hgdp/hgdp_structural_variation). We merged long-read and short-read SV sets using parameters from *Aganezov et al., 2020* (maximum distance of 500 bp between SV start sites, maximum difference of 10% between SV lengths). We chose not to require matching on SV type, strand orientation, or insertion sequence due to inconsistencies in SV representations across the short-read and long-read variant callers (*Aganezov et al., 2020*). In order to match the minimum SV length used by the short-read datasets, we also removed all long-read discovered SVs shorter than 50 bp. For the purpose of broad allele frequency binning for SVs called in each of these sets, we then calculated dataset-wide allele frequencies with PLINK v1.90b6.4 (*Purcell et al., 2007*).

## Graph genotyping of structural variation

High-coverage 30× short-read sequencing data for the core 2504 individuals in 1KGP, sequenced by the New York Genome Center (*Byrska-Bishop et al., 2021*), was obtained from ENA (PRJEB31736). We genotyped SVs in these samples with Paragraph v2.2 (*Chen et al., 2019*). In accordance with Paragraph's recommendations, we set the maximum permitted read count for variants to 20 times the mean sample depth in order to limit runtime for repetitive regions. Genotypes from all samples were combined using bcftools v1.9 (*Danecek et al., 2021*).

To obtain a high-quality set of genotyped SVs, we filtered the resulting data based on dataset-wide genotyping rates and within-population Hardy–Weinberg equilibrium. We determined an SV's overall genotyping rate with cyvcf2 (*Pedersen and Quinlan, 2017*) and removed variants that were not genotyped in ≥50 % of samples. We additionally calculated one-sided Hardy–Weinberg equilibrium p-values (excess of heterozygotes) for variants within each of the 26 1KGP populations, using the Hardy–Weinberg package from R (*Graffelman, 2015*). We filtered out SVs that violated equilibrium expectations (Fisher's exact test, $p < 1 \times 10^{-4}$) in ≥13 populations. Unfolded, within-population allele frequencies were calculated with PLINK.

## Quantifying linkage disequilibrium with known SNPs and short indels

To calculate linkage disequilibrium (LD) between SVs and SNPs or short indels in 1KGP samples, we used small variant genotypes produced by the 1000 Genomes Consortium. These genotypes were generated by aligning the 1KGP Phase 3 data to GRCh38 and then calling variants against the GRCh38 reference and are restricted to biallelic SNVs and indels (http://ftp.1000genomes.ebi.ac.uk/vol1/ftp/data_collections/1000_genomes_project/release/20190312_biallelic_SNV_and_INDEL/). Y chromosome genotypes, not included in the former data release, were obtained from Phase 3 variant calls lifted over to GRCh38 (http://ftp.1000genomes.ebi.ac.uk/vol1/ftp/release/20130502/supporting/GRCh38_positions/). We combined the 1KGP SNP and indel genotypes with our SV genotypes and calculated $r^2$ within each population, using a 100-variant and 100 Mb window, with PLINK v1.90b6.4.

Genome accessibility masks from 1KGP were obtained from http://ftp.1000genomes.ebi.ac.uk/vol1/ftp/data_collections/1000_genomes_project/working/20160622_genome_mask_GRCh38/, and ENCODE blacklisted regions were obtained from https://www.encodeproject.org/files/ENCFF356LFX/.

We note peaks of $r^2$ around discrete values of 1, 0.5, 0.33, 0.25, etc. (*Figure 1—figure supplement 7*), which occur due to the properties of the $r^2$ statistic for various allele frequency combinations. Specifically, the maximum possible value of $r^2$ for a pair of rare alleles with frequencies $p_a$ and $p_b$ is described by *VanLiere and Rosenberg, 2008* as:

$$r^2_{max}\left(p_a, p_b\right) = \frac{p_a\left(1-p_b\right)}{\left(1-p_a\right)p_b}$$

Assuming a sample size of 198 chromosomes (the most common sample size for a given 1KGP population), the maximum possible value of $r^2$ for a pair of singletons is therefore:

$$p_a = 1/198$$

$$p_b = 1/198$$

$$r^2_{max} = 1$$

Meanwhile, the maximum possible value of $r^2$ for a singleton-doubleton pair is:

$$p_a = 1/198$$

$$p_b = 2/198$$

$$r^2_{max} = 0.497$$

And the maximum possible value of a singleton-tripleton pair is:

$$p_a = 1/198$$

$$p_b = 3/198$$

$$r^2_{max} = 0.330$$

Given that the most common allele frequency of a SNP or SV is 1/N (i.e., singletons), followed by 2/N (doubletons), 3/N (tripletons), etc., the aforementioned allele frequency combinations occur frequently, resulting in some common discrete values of the maximum $r^2$.

## eQTL mapping

To conduct eQTL analysis, we used gene expression data generated by the Geuvadis Consortium (*Lappalainen et al., 2013*), which includes 447 intersecting samples from the following 1KGP populations: CEU, FIN, GBR, TSI, and YRI. Using the recount3 package from R/Bioconductor (*Collado-Torres et al., 2017*), we extracted gene expression counts for all corresponding samples. Counts were normalized across samples using the trimmed mean of M values (TMM) method from edgeR

(*Robinson et al., 2010*). TPM values were also computed from raw counts. In accordance with the methods employed for eQTL mapping in the GTEx Project (*The GTEx Consortium, 2020*), we retained all genes with TPM values greater than or equal to 0.1, as well as raw read counts greater than or equal to six in at least 20% of samples. We then applied rank normalization to the TMM values for each remaining gene. We then performed *cis*-eQTL mapping with a modified version of fastQTL (https://github.com/hall-lab/fastqtl, *Colby, 2016*, *Ongen et al., 2016*), which accounts for SV size when determining the appropriate *cis* window. We conducted nominal and permutation passes with genotype principal components and sex included as covariates. Beta-distribution-approximated permutation p-values from fastQTL were used as input to estimate q-values and control the false discovery rate (FDR) with the qvalue package (http://github.com/jdstorey/qvalue, *Storey et al., 2020*, *Storey and Tibshirani, 2003*).

Next, we used the ChromHMM epigenetic state annotations generated by the Roadmap Epigenomics Project (*Roadmap Epigenomics Consortium et al., 2015*) for the E116 B lymphocyte cell line to determine whether SVs eQTLs were enriched for regulatory elements. For each of the 15 ChromHMM epigenetic states, we identified the SVs that overlapped with its epigenetic state annotations and determined its enrichment for SV eQTLs using Fisher's exact test.

To identify potential causal variants for the 1121 significant gene-SV pairs, we performed eQTL fine-mapping using CAVIAR v2.2 (*Hormozdiari et al., 2014*). Nominal pass output of fastQTL was combined for SNPs and SVs, and p-values were back-converted to z-scores while also accounting for the sign of the association (i.e., positive or negative). For each significant eQTL, we extracted all tested variants within a 1 Mb window, excluding any variants with a nominal p-value > 0.05. We then used PLINK to calculate the LD matrix relating this set of variants. Using these two inputs, we then ran CAVIAR for each of the 1121 significant gene-SV pairs, recording the 90% credible causal set for each eQTL.

## Admixture-aware scan for signatures of local adaptation

Common examples of frequency differentiation-based metrics to search for selection include Wright's fixation index ($F_{ST}$) (*Wright, 1949*), as well as tree-based extensions of this concept such as the locus-specific branch length (LSBL) (*Shriver et al., 2004*) and population branch statistic (PBS) (*Yi et al., 2010*). While useful for polarizing frequency changes on particular lineages, these tree-based tests still require the specification of (typically three) populations for comparison. The number of possible comparisons thus grows in a combinatorial manner with the number of populations in the study. Specifically, for the 26 populations of 1KGP, there are 2600 (26 choose 3) possible comparisons. A second limitation of such tests is the definition of population, which may or may not reflect genetic patterns of population structure that occur at multiple scales. Moreover, many human populations exhibit substantial admixture, which is ignored by, and may confound, some frequency differentiation-based tests.

We overcame these limitations by using the software package Ohana (*Cheng et al., 2019*) to scan SVs for signatures of positive selection, that is extreme frequency differentiation between populations. Ohana uses allele frequency to model individuals as an admixed combination of *k* ancestral populations, constructing a genome-wide covariance matrix to describe the relationship between these ancestry components. Variants are then assessed to determine whether their allele frequencies are better explained by the genome-wide 'neutral' matrix, or by an alternative matrix that allows allele frequencies to vary in one ancestry component.

In accordance with Ohana's recommended workflow, we conducted admixture inference on the 1KGP dataset with a set of ~100,000 variants, downsampled from Chromosome 21 of the 1KGP SNV/indel callset used for LD calculations above. Downsampling was performed with PLINK's variant pruning function. Inference of admixture proportions in the dataset was allowed to continue until increased iterations produced qualitatively similar results (50 iterations). The covariance matrix generated from this downsampled dataset was used as a neutral input for downstream scans for selection on SVs.

In order to generate 'selection hypothesis' matrices to search for selection in a specific ancestry component, we modified the neutral covariance matrix by allowing one component at a time to have a greater covariance. A scalar value of 10, representing the furthest possible deviation that a variant could have in a population under selection, was added to elements of the neutral covariance

matrix depending on the population of interest. For each variant of interest, Ohana then computes the likelihood of the observed ancestry-component-specific allele frequencies under the selection and neutral models, then compares them by computing a likelihood ratio statistic (LRS), which quantifies relative support for the selection hypothesis. We note that the 'selection hypothesis' matrices we generated for our study enable us to perform tests for frequency differentiation in one ancestry component at a time. Testing for selection in multiple ancestry components would require modifying multiple components in the neutral covariance matrix. While some variants have outlier LRS values in multiple ancestry components, this reflects a combined signature of extremely high allele frequency in one ancestry component and extremely low allele frequency in another (e.g., the *IGHG4* insertion we highlight is an outlier in both ancestry component 2, where it reaches high allele frequencies, and ancestry component 6, where it is much rarer than expected from the null model [*Figure 3B*, panel for ancestry component 6]).

We filtered our results to remove extreme outliers in null model log-likelihoods (global log likelihood [LLE] < −1000), which were unremarkable in their patterns of allele frequency and instead indicated a failure of the neutral model to converge for a small subset of rare variants. To calculate p-values, we then compared the LRS to a chi-square distribution with one degree of freedom (*Wojcik et al., 2019*) and adjusted for multiple hypothesis testing using a Bonferroni correction. To further refine the list of candidate selected loci, we also compared the ancestry component-specific LRS computed for each SV to the observed distribution of LRS computed from SNPs and short indels matched on global minor allele frequency (in 1% frequency bins). Specifically, we identified SVs with LRS exceeding the 99.9 percentile of the empirical LRS distribution for frequency-matched SNPs and short indels as calculated using identical methods. These background SNPs and indels were limited to Chromosome 1 for computational efficiency, but results were qualitatively unaffected by the choice of chromosome.

## Assessment of archaic introgression

To identify candidate archaic introgressed SVs among the set of highly differentiated variants, we tested each SV for LD with putative introgressed SNPs as identified based on published results from the method Sprime, which is based on signatures of LD and divergence from an African outgroup population (*Browning et al., 2018*). Specifically, we computed pairwise LD between the SV and all SNPs in a 100 kb window, matching the ancestry component to a corresponding population from 1KGP. We additionally computed the allele frequencies of SVs in African populations, excluding the ASW (Americans of African Ancestry in SW USA) and ACB (African Caribbeans in Barbados) populations which exhibit substantial non-African admixture. We reported all highly differentiated SVs possessing $r^2 > 0.5$ with any putative introgressed SNP and AF < 0.01 in non-admixed African populations (*Supplementary file 2*).

We obtained masks for the Altai Neanderthal sequence, which were used in filtering by *Browning et al., 2018*, from http://cdna.eva.mpg.de/neandertal/Vindija/FilterBed/. These masked regions filter for coverage (stratified by GC content), require a minimum coverage of 10× , require all 35-mers in the region to be unique, and exclude tandem repeats and indels.

To efficiently search for the *IGHG4* insertion in additional datasets, we designed a 48 bp sequence (TGGAGAGAGTGGGGGACAGCGTCAGGGACAGGTGGGGACAGCCTGGGG) that spans the insertion breakpoint, extending across the entire 33 bp of the insertion itself as well as 11 bp and 4 bp, into the respective upstream and downstream flanking regions. BLAST searches for this sequence in the hg19 and GRCh38 human reference genomes returned no exact matches, but the diagnostic sequence was sufficiently similar to the reference sequence that reads containing it still mapped to the same locus. Sequence alignments from four high-coverage archaic hominin samples (Altai Neanderthal, Vindija Neanderthal, Chagyrskaya Neanderthal, and Denisovan) were obtained from http://cdna.eva.mpg.de/neandertal and http://cdna.eva.mpg.de/denisova. Forward strand sequences of unique reads were extracted from reads aligned to the hg19 reference genome, and exact matches to the 48 bp query sequence were identified using grep.

Evidence of introgression at the *IGH* locus was further examined by counting observed alleles at highly differentiated SNPs from sequenced alignments for each high-coverage archaic sample (see above). Sites with two or more reads supporting the alternative allele were used to define matching

and color *Figure 4*. further conditions on alternative allele frequency ≤ 1 % within African populations of 1KGP.

## Inference of selection parameters with approximate Bayesian computation

We used a sequential algorithm for approximate Bayesian computation (*Lenormand et al., 2013*; *Pritchard et al., 1999*), implemented with the R package EasyABC (*Jabot et al., 2013*), to infer the strength and timing of selection at the *IGHG4* locus, as well as the initial frequency of the adaptive allele. This approach consisted of drawing model parameters from prior distributions as input to the forward evolutionary simulation software package SLiM (*Haller and Messer, 2019*), computing summary statistics from each simulation, and comparing to those observed from our data. Simulations with summary statistics most closely matching the observed data are then used to construct posterior distributions of the model parameters. The sequential algorithm automatically determines the tolerance level and uses a predetermined stopping criterion ($p_{accmin}$ = 0.05), thereby reducing the necessary number of simulations and improving estimates of the posterior distribution (*Lenormand et al., 2013*).

We constructed a simplified five-population demographic model based on parameters obtained from *Jouganous et al., 2017* and *Wang et al., 2018*, as well as population size estimates from 1KGP (*The 1000 Genomes Project Consortium, 2015*; *Figure 5C*). Specifically, our model consisted of an initial divergence event between the CEU and East Asian populations at 46 kya, subsequent divergence between the population ancestral to CHB/JPT and the population ancestral to KHV/CHX at 9.8 kya, and final splits between CHB/JPT and KHV/CHX at 9.0 kya and 1.7 kya, respectively. Each new population was drawn from its originating population at the same size of the originating population. The size of the ancestral population was set at 2831 (*Jouganous et al., 2017*) and allowed to expand exponentially at a rate of $1.25 \times 10^{-3}$ per generation, resulting in a final size of 17,883 in each subpopulation, broadly consistent with pairwise sequential Markovian coalescent (PSMC) results presented by 1KGP for these populations (*The 1000 Genomes Project Consortium, 2015*). We allowed the initial allele frequency of the selected variant ($p_0$) to vary between 0 and 1, the timing of the onset of selection ($T_{adaptive}$) to vary between 1 and 1686 generations ago (i.e., the entire simulated timespan), and the selection coefficient ($s$) to vary between –0.01 and 0.2 – all drawn from uniform distributions. We used the selected variant's end-of-simulation allele frequencies as a summary statistic for the model, comparing these frequencies to the observed allele frequencies of the *IGHG4* insertion in these five populations.

To obtain a background distribution of the expected frequencies of the introgressed haplotype in the absence of selection, we performed 100,000 of the simulations described above with a selection coefficient of 0. For ease of comparison to the observed data, we used the end-of-simulation population branch statistic (PBS) (*Yi et al., 2010*), calculated between the CDX, JPT, and CEU populations, as a summary statistic. We compared these PBS values, as well as the inter-population $F_{ST}$, to those observed for the *IGHG4* insertion in these populations.

## Phenotype-wide association analysis

We examined potential phenotype associations with the putative Neanderthal-introgressed haplotype at the *IGH* locus by extracting summary statistics from the pan-ancestry analysis of the UK Biobank (https://pan.ukbb.broadinstitute.org/). Specifically, we obtained association p-values for two SNPs, which each tag one of the two major LD blocks (rs115091999 and rs150526114). We restricted analysis to individuals of East Asian ancestry. No variants were significant after Bonferroni correction (*Figure 5—figure supplement 2*).

## Acknowledgements

We thank Tim O'Connor, Sai Chen, John Kim, and members of the McCoy lab for feedback and helpful discussions. We also thank the staff at the Maryland Advanced Research Computing Center for computing support. This work is supported by the National Institutes of Health (NIH) grant R35GM133747 to RCM and the US National Science Foundation grant DBI-1350041 to MCS.

## Additional information

### Funding

| Funder | Grant reference number | Author |
|---|---|---|
| National Institutes of Health | R35GM133747 | Rajiv C McCoy |
| National Science Foundation | DBI-1350041 | Michael C Schatz |

The funders had no role in study design, data collection and interpretation, or the decision to submit the work for publication.

### Author contributions

Stephanie M Yan, Conceptualization, Formal analysis, Investigation, Methodology, Software, Visualization, Writing – original draft, Writing – review and editing; Rachel M Sherman, Data curation, Investigation, Software, Writing – review and editing; Dylan J Taylor, Formal analysis, Investigation, Methodology, Software, Visualization, Writing – review and editing; Divya R Nair, Investigation, Software; Andrew N Bortvin, Investigation, Software, Visualization; Michael C Schatz, Conceptualization, Funding acquisition, Writing – review and editing; Rajiv C McCoy, Conceptualization, Formal analysis, Funding acquisition, Investigation, Methodology, Software, Visualization, Writing – original draft, Writing – review and editing

### Author ORCIDs

Stephanie M Yan http://orcid.org/0000-0002-6880-465X
Rachel M Sherman http://orcid.org/0000-0003-1750-8428
Dylan J Taylor http://orcid.org/0000-0001-5806-4494
Divya R Nair http://orcid.org/0000-0002-3374-8424
Andrew N Bortvin http://orcid.org/0000-0001-8784-6786
Michael C Schatz http://orcid.org/0000-0002-4118-4446
Rajiv C McCoy http://orcid.org/0000-0003-0615-146X

### Decision letter and Author response

Decision letter https://doi.org/10.7554/eLife.67615.sa1
Author response https://doi.org/10.7554/eLife.67615.sa2

## Additional files

### Supplementary files

• Supplementary file 1. Highly differentiated SV loci across ancestry components. The top three SVs per ancestry component are reported. The *IGH* insertion and deletion are highlighted in bold text.

• Supplementary file 2. Highly differentiated SVs that are also significant eQTLs.

• Supplementary file 3. Highly differentiated SVs in LD ($r^2$ >0.5) with putative archaic introgressed haplotypes called by Sprime. The *IGH* insertion and deletion are highlighted in bold text.

• Transparent reporting form

### Data availability

All code necessary for reproducing our analysis is available on GitHub (https://github.com/mccoy-lab/sv_selection; copy archivedat https://archive.softwareheritage.org/swh:1:rev:2e105bf050fd2acb9f1e0d1833566c0b1ca5b41f). SV genotypes, eQTL results, and selection scan results are available on Zenodo (https://doi.org/10.5281/zenodo.5509980).

The following dataset was generated:

| Author(s) | Year | Dataset title | Dataset URL | Database and Identifier |
|---|---|---|---|---|
| Yan SM, McCoy RC | 2021 | Data from: Local adaptation and archaic introgression shape global diversity at human structural variant loci | https://doi.org/10.5281/zenodo.5509980 | Zenodo, 10.5281/zenodo.5509980 |

The following previously published datasets were used:

| Author(s) | Year | Dataset title | Dataset URL | Database and Identifier |
|---|---|---|---|---|
| Audano PA | 2019 | Long-read sequencing of a Puerto Rican individual (HG00733) | https://www.ncbi.nlm.nih.gov/bioproject/300840 | NCBI BioProject, PRJNA300840 |
| Audano PA | 2019 | Long-read sequencing of a Yoruban individual (NA19240) | https://www.ncbi.nlm.nih.gov/bioproject/288807 | NCBI BioProject, PRJNA288807 |
| Audano PA | 2019 | Long-read sequencing of a Gambian individual (HG02818) | https://www.ncbi.nlm.nih.gov/bioproject/339722 | NCBI BioProject, PRJNA339722 |
| Audano PA | 2019 | Long-read sequencing of a Luhya individual (NA19434) | https://www.ncbi.nlm.nih.gov/bioproject/385272 | NCBI BioProject, PRJNA385272 |
| Audano PA | 2019 | Long-read sequencing of a Vietnamese individual (HG02059) | https://www.ncbi.nlm.nih.gov/bioproject/339726 | NCBI BioProject, PRJNA339726 |
| Audano PA | 2019 | Long-read sequencing of a Utah individual with Northern and Western European ancestry (NA12878) | https://www.ncbi.nlm.nih.gov/bioproject/323611 | NCBI BioProject, PRJNA323611 |
| Audano PA | 2019 | Long-read sequencing of a Telugu individual (HG04217) | https://www.ncbi.nlm.nih.gov/bioproject/481794 | NCBI BioProject, PRJNA481794 |
| Audano PA | 2019 | Long-read sequencing of a Peruvian individual (HG02106) | https://www.ncbi.nlm.nih.gov/bioproject/480858 | NCBI BioProject, PRJNA480858 |
| Audano PA | 2019 | Long-read sequencing of a Finnish individual (HG00268) | https://www.ncbi.nlm.nih.gov/bioproject/480712 | NCBI BioProject, PRJNA480712 |
| Audano PA | 2019 | Long-read sequencing of a Han Chinese individual (HG00514) | https://www.ncbi.nlm.nih.gov/bioproject/PRJNA300843 | NCBI BioProject, PRJNA300843 |
| Byrska-Bishop M | 2019 | 1000 Genomes Project phase 3: 30X coverage whole genome sequencing | https://www.ncbi.nlm.nih.gov/bioproject/527456 | NCBI BioProject, PRJEB31736 |
| Huddleston J | 2017 | Long-read sequencing of a hydatidiform mole (CHM1) | https://www.ncbi.nlm.nih.gov/bioproject/246220 | NCBI BioProject, PRJNA246220 |
| Huddleston J | 2017 | Long-read sequencing of a hydatidiform mole (CHM13) | https://www.ncbi.nlm.nih.gov/bioproject/269593 | NCBI BioProject, PRJNA269593 |
| Lappalainen T | 2013 | RNA-sequencing of 465 lymphoblastoid cell lines from the 1000 Genomes | https://www.ebi.ac.uk/arrayexpress/experiments/E-GEUV-1/ | ArrayExpress, E-GEUV-1 |

*Continued on next page*

*Continued*

| Author(s) | Year | Dataset title | Dataset URL | Database and Identifier |
|---|---|---|---|---|
| Seo J-S | 2016 | Long-read sequencing of a Korean individual (AK1) | https://www.ncbi.nlm.nih.gov/bioproject/298944 | NCBI BioProject, PRJNA298944 |
| Shi L | 2016 | Long-read sequencing of a Chinese individual (HX1) | https://www.ncbi.nlm.nih.gov/bioproject/301527 | NCBI BioProject, PRJNA301527 |

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
