## [Decision Letter]

**Acceptance summary:**

The technical challenges of identifying and quantifying the frequency of structural variants (SV) on a population scale has been a major limitation to the study of recent human adaptation. This manuscript applies a recent graph-based genotyping method that leverages a library of SVs identified by long-read sequencing to identify SVs in large short-read based cohorts. This is a sensible and powerful approach that highlights several examples of likely adaptive SV evolution in different human populations.

**Decision letter after peer review:**

Thank you for submitting your article "Local adaptation and archaic introgression shape global diversity at human structural variant loci" for consideration by *eLife*. Your article has been reviewed by 3 peer reviewers, one of whom is a member of our Board of Reviewing Editors, and the evaluation has been overseen by George Perry as the Senior Editor. The reviewers have opted to remain anonymous.

Essential revisions:

1) Clearer connection and contextualization of the contribution of this work given previous studies, both for the SV catalog of results and the highlighted IGH, including analyses of overlap with previous datasets and a clearer set up to enable this paper to serve a larger purpose as a roadmap for future studies that aim to link datasets for SVs. Related to this, the language in a number of sections is unclear or too broad to scientifically interpret, e.g. suggesting analyses are broadly consistent with previous work; specific quantitative comparisons are needed to make or interpret these claims.

2) More depth to the biological interpretation of the study, with specific ideas presented below, including further support/interpretation of the IGH locus and potential tests of SV patterns more broadly.

*Reviewer #1 (Recommendations for the authors):*

More minor or specific comments:

1) The populations and ancestries are sometimes hard to follow with so many population abbreviations and the 8 ancestries in Ohana figures not quite linked to a populations. e.g. What ancestry backgrounds are the SVs of interest on pg 14?

2) In general, while clear to specialists, early in the paper the writing is difficult, with many stats tests before the biological question you are attempting to test.

3) ABC demography model is quite complex; does this matter for the inference? Importantly, how does it mix with Neanderthal introgression scenario? It is also unclear what summary statistics are being used in the ABC model, is it the Ohana/PBS-style only? If other measures, are they admixture-aware? The point estimate of "strong" selection is hard to trust with so much model uncertainty and identifiability issues with timing.

4) Further support for the claim of selection would strengthen the argument, i.e. other summary statistics that show signatures of selection.

*Reviewer #2 (Recommendations for the authors):*

1. The authors do a good job motivating the importance of SVs in human evolution and the previous technical limitations that have limited comprehensive analysis. They also clearly motivate their work using recently generated population-level SV datasets. However, clarification of how this paper's results compare to and expand these previous findings is needed. Previous analyses of SV evolution were not "comprehensive" due to the limitations of short-read approaches. However, as noted by the authors, this and other long-read-based studies also currently cannot "comprehensively" study human SV. I do not see a problem with the "focus on individual variants" obtained in a different way, but more description of how their set of SVs is similar or different from previous work would aid interpretation. A few more supplementary figures describing characteristics of the 92,286 SVs used for downstream analysis (length, type of SV, genome distribution, SV overlap, frequency, etc) are needed. Even though they are already cited, some papers the authors could consider further contextualizing their results in include: Hsieh et al., 2019., Sudmant et al., 2015, Almarri et al., 2020, Ebert et al., 2020, Audano et al., 2019 (there are also others that consider SV with direct reference to introgression events).

2. Related to the above point, the use of the graph genotyping software, Paragraph, is s sensible approach, and I suspect that this paper may provide a template for future analyses merging long-read and short-read SV data. Given this potential contribution, more should be done to discuss the benefits and limitations of this approach. There is brief discussion of "ascertainment bias" based on the 15 individuals with long-read data used from Audano et al., 2019. How does the ancestry of these 15 individuals influence the ability to comprehensively identify SVs in thousands of individuals with different ancestries? A few more details in the main text regarding the sensitivity and specificity of Paragraph would also help to aid result interpretation (rather than just "recent benchmarking…support the accuracy of Paragraph"). Does the Paragraph approach better identify longer SVs, deletions, duplications, inversions, SVs in more repetitive regions of the genome?

3. The authors identify 1121 significant associations between SVs and the expression of nearby genes in LCLs. This analysis is reasonable and clear, but the implications are not clear. Prior to this section, the authors discuss LD with previously investigated SNPs/indels, yet they do not report this information for the eQTL 1121 SVs. How many of these 1121 are in high LD with common SNPs/indels? And how many are not linked to previously investigated variation? Even if most are linked, this is still potentially interesting, but it would be helpful to know if they are identified as eQTL in other datasets (GTEx?). Do these potentially SV mediated eQTL have greater effect sizes than eQTL caused by SNVs? Is there a way to fine map one or two (potentially one related to the subsequently immunoglobulin story or the ones highlighted in Figure 2C) to show that the SV is likely driving the association, rather than other linked variants?

4. The authors use Ohana to test for local adaptation. This method seems appropriate for the challenges of the SV data; the results are interesting; and I appreciate their new individual examples. However, these results are largely descriptive. Testing larger hypotheses about whether SVs are targets of selection more often than SNVs or if these patterns differ between populations or parts of the genome would substantially increase the impact of these results. Figure 4-S1 may be relevant to this kind of comparison; however, it only shows a few loci. While differences in power and ascertainment may make direct comparison challenging, I encourage the authors to think about how to move beyond a list of examples to more general conclusions. Additionally, the y-axis range on Figure 4 across the different ancestries are quite different. It is hard to interpret to the significance and magnitude of the LRS. Can the multiple testing control be integrated here?

5. Finally, the authors explore one example, the immunoglobulin heavy chain locus, in detail. The example and analysis are compelling; however, it is complicated and sometimes hard to follow given the multiple SVs in this region. Readers would benefit from a cartoon schematic of this locus, perhaps expanding on the supplemental figure. I would also appreciate more discussion of why this region might have been filtered out in Browning et al., 2018.

*Reviewer #3 (Recommendations for the authors):*

The abstract and introduction focus on the size of SVs and the relative "ease" of analysis of SNVs and indels, however, the highlighted variants are indels by the 1000 genomes definition. They also seem to have been discovered in other short-read datasets, (e.g. gnomAD).

Paragraph has been previously shown to work well for genotyping SVs, however it is not clear what the accuracy in the low-coverage 1000 genomes samples is. After filtering, while 92,286 variants remain, 25,201 (27%) of these are absent from any of the 1000 genomes samples. It is unclear how the authors distinguish "true negatives" from "false negatives."

I don't think the LD section adds anything new. This is well reported.

And LD cutoff of 0.5 R^2^ is used for introgressed haplotypes versus 0.8^2^ for tag haplotypes in the original motivation. Suggest picking one to be consistent throughout.

---

## [Author Response]

Essential revisions:1) Clearer connection and contextualization of the contribution of this work given previous studies, both for the SV catalog of results and the highlighted IGH, including analyses of overlap with previous datasets and a clearer set up to enable this paper to serve a larger purpose as a roadmap for future studies that aim to link datasets for SVs. Related to this, the language in a number of sections is unclear or too broad to scientifically interpret, e.g. suggesting analyses are broadly consistent with previous work; specific quantitative comparisons are needed to make or interpret these claims.

Thank you very much for the thorough and thoughtful reviews, which were invaluable for improving our manuscript. We agree that the previous draft placed too much responsibility on readers to extract and compile information from cited studies. We made several changes with the goals of placing our work in context and offering clear, quantitative comparisons:

1. One of the main claims of our manuscript is the distinction between short-read- and long-read-discovered SVs, with the statement that the latter set is more accurate and comprehensive. Our set of SVs was carefully curated based on PacBio long-read sequencing data from 15 diverse samples from Audano et al., (2019). We now provide a detailed comparison of these curated SVs to two sets of SVs discovered from short-read sequencing of thousands of globally diverse human samples from the 1000 Genomes Project and Human Genome Diversity Project (HGDP) (Almarri et al., 2020; Sudmant et al., 2015) (lines 80-97; Figure 1 – S2, Figure 1 – S3). We show that we are able to discover significantly more insertions with long-read sequencing and re-discover a large proportion of common SVs despite the small size of the long-read-sequenced discovery sample.

2. Another claim of our study regards the accuracy of graph genotyping with Paragraph. Our focus on eQTL mapping and selection scans offers some inherent robustness to genotyping errors, because in order for false positive associations to arise, erroneous genotypes would have to be unequally distributed with respect to gene expression phenotypes or ancestry. This is implausible unless driven by stratified genetic variation elsewhere in the genome (i.e., by true positives, albeit mis-attributed). False negatives, meanwhile, would make our eQTL mapping and selection scans conservative. We now provide additional results supporting the accuracy of graph genotyping on lines 100-107, including important details of previous benchmarking studies.

3. We also include a more detailed comparison of our introgression results to those of Browning et al., on lines 407-417. The Browning et al., study highlights a subregion of the IGH locus, which reaches high frequencies in European and Asian populations and is downstream of the region we investigate. This is displayed in Figure 4 – S3, where the Browning et al., haplotype is represented in the EUR, EAS, and AMR populations, while the one we investigate is only in the EAS populations (with the exception of JPT). The precise reason why Browning et al., focused on the Eurasian haplotype, rather than the one at high frequency in southeast Asia alone, is unclear. We hypothesize that this may involve their filtering process, which used a mask for the Altai Neanderthal sequence, generated by (Prüfer et al., 2017), that removes sites that have low coverage or mapping quality or are located within tandem repeats or indels (Browning et al., 2018). We now add a supplementary figure showing the masking of the broader IGH region that we highlight (Figure 4 – S4), where only 19.3% of the sequence remains after masking (in contrast to the background rate of 62.6% across the remainder of the q-arm of chromosome 14).

2) More depth to the biological interpretation of the study, with specific ideas presented below, including further support/interpretation of the IGH locus and potential tests of SV patterns more broadly.

1. We added more biological context to our eQTL analysis by intersecting SVs with annotated regulatory elements (lines 215-220). We observed a strong enrichment of significant gene expression associations for SVs that intersect annotated enhancer elements, as well as a strong depletion of expression associations for SVs that intersect “quiescent” sequences that are devoid of important epigenetic marks (Figure 2D). The intuitive nature of these results supports the robustness of our analysis, while also shedding light on functional mechanisms through which SVs mediate impacts on phenotypes and fitness.

2. We investigated direct dosage effects of exon-spanning SVs, showing that the magnitude and direction of these effects are broadly consistent with expectations (lines 220-225). We also performed a fine-mapping analysis of SV and SNP eQTLs (lines 228-244), homing in on candidate causal variants at SV eQTL loci. While noting important technical caveats, both of these analyses clarify the causal link between a subset of SVs and the observed gene expression changes.

3. We additionally identified seven frequency-differentiated SVs from our study that were also significant eQTLs (Supplementary Table 2). One major limitation of this analysis is the lack of diversity in the Geuvadis dataset, which restricts us to SVs that are common in European or Yoruban populations. Consequently, we were unable to directly test gene expression associations for highlighted SVs at the *IGH* locus in East Asian samples, because this ancestry group is not represented in Geuvadis.

4. We performed extensive neutral simulations to construct null distributions of test statistics and strongly support our conclusion of positive selection at the IGH locus in the ancestors of certain southeast Asian populations. Specifically, observed magnitudes of allele frequency differentiation far exceed those expected under the neutral model (lines 494-496; Figure 5 – S1).

Please find our point-by-point response to the reviewers appended below.

Reviewer #1 (Recommendations for the authors):More minor or specific comments:1) The populations and ancestries are sometimes hard to follow with so many population abbreviations and the 8 ancestries in Ohana figures not quite linked to a populations. e.g. What ancestry backgrounds are the SVs of interest on pg 14?

To address this comment, which was shared with that of another reviewer, we have combined Figures 3A and 4 into a revised Figure 3, such that ancestry patterns for the 1000 Genomes populations are now shown along with selection results for each of the 8 ancestry components.

2) In general, while clear to specialists, early in the paper the writing is difficult, with many stats tests before the biological question you are attempting to test.

Thank you for this suggestion. We have attempted to address this by adding clear topic sentences and restructuring several of these initial paragraphs to describe the motivating questions before describing the results.

3) ABC demography model is quite complex; does this matter for the inference? Importantly, how does it mix with Neanderthal introgression scenario? It is also unclear what summary statistics are being used in the ABC model, is it the Ohana/PBS-style only? If other measures, are they admixture-aware? The point estimate of "strong" selection is hard to trust with so much model uncertainty and identifiability issues with timing.

Thank you for pointing out these concerns. We agree that our model is fairly complex and relies on estimates of demographic parameters from the literature. We chose to incorporate five populations in order to have the resolution for capturing differences in the allele frequency of the *IGHG4* insertion in East Asia (i.e., near fixation in the CDX and KHV Southeast Asian populations, intermediate frequency in CHB, almost absent in JPT, and CEU as an outgroup). Because the estimates of population divergence time and size used in our model come from two sources (Jouganous et al., 2017; Wang et al., 2018), and are broadly consistent between these two methods, we believe that the model is sufficiently accurate to provide a coarse estimate of selection parameters at this locus. We now include a more in-depth discussion of these caveats in the discussion (lines 583-586).

We apologize that the previous description of ABC summary statistics was unclear. We used the end-of-simulation allele frequencies of the simulated variant in each of the five populations, and compared them to the known allele frequencies of the *IGHG4* insertion in these populations. We clarify this choice in the Methods (lines 803-805).

4) Further support for the claim of selection would strengthen the argument, i.e. other summary statistics that show signatures of selection.

We currently only evaluate selection at the *IGH* locus using methods based on allele frequencies. We are hesitant to incorporate haplotype-based summary statistics due to the known challenge of genotyping and phasing within the broader immunoglobulin region (outside of the *IGH* genes), which exhibits high levels of genetic variation exacerbated by the phenomena of somatic recombination and hypermutation that occurs in developing B-cells (Chin et al., 2020; Zhang et al., 2021).

We now provide further support for our conclusion of selection at this locus by performing 100,000 neutral simulations using the 5-population demographic model based on (Jouganous et al., 2017; Wang et al., 2018). We show that the magnitudes of allele frequency differences, quantified as pairwise frequency differences as well as with the population branch statistic, greatly exceed those expected based on genetic drift alone (lines 494-496; Fig. 5 - S1). This approach is similar to that employed in previous literature (e.g., Ilardo et al., 2018). The omission of migration from the simulations makes our conclusion conservative, in that gene flow would act to reduce allele frequency differences between populations.

Reviewer #2 (Recommendations for the authors):1. The authors do a good job motivating the importance of SVs in human evolution and the previous technical limitations that have limited comprehensive analysis. They also clearly motivate their work using recently generated population-level SV datasets. However, clarification of how this paper's results compare to and expand these previous findings is needed. Previous analyses of SV evolution were not "comprehensive" due to the limitations of short-read approaches. However, as noted by the authors, this and other long-read-based studies also currently cannot "comprehensively" study human SV. I do not see a problem with the "focus on individual variants" obtained in a different way, but more description of how their set of SVs is similar or different from previous work would aid interpretation. A few more supplementary figures describing characteristics of the 92,286 SVs used for downstream analysis (length, type of SV, genome distribution, SV overlap, frequency, etc) are needed. Even though they are already cited, some papers the authors could consider further contextualizing their results in include: Hsieh et al., 2019., Sudmant et al., 2015, Almarri et al., 2020, Ebert et al., 2020, Audano et al., 2019 (there are also others that consider SV with direct reference to introgression events).

Thank you very much for these suggestions, which are closely related to comments by Reviewer 1. We now provide a detailed comparison of these curated SVs to two sets of SVs discovered from short-read sequencing of diverse human samples (Almarri et al., 2020; Sudmant et al., 2015) (lines 80-97). We find that this long-read-discovered SV set includes 89,979 variants (83.4% of long-read SVs) that are not represented in the 1000 Genomes Project (1KGP) or the Human Genome Diversity Project (HGDP). These long-read specific variants include 30,229 that are “common” (AF ≥ 0.05), representing 72.3% of all common SVs. We were also able to rediscover a large proportion of the short-read-discovered SVs in these two datasets, including 66.0% and 17.7% of common SVs in 1KGP and HGDP, respectively (Figure 1 – S2 and Figure 1 – S3). These results are consistent with reports from previous studies (Zhao et al., 2021).

The overlap we describe above is notable given that the much smaller size of the long-read sample set (15 individuals vs. 2,504 for 1KGP and 911 for HGDP), and that the sample sets do not overlap completely (i.e., we expect that many rare or singleton SVs should not be represented in both datasets). We expect that the SVs unique to the short-read datasets reflect both differences in the discovery sample set (i.e., many of the long-read sequenced individuals are also in 1KGP, while none are in HGDP) and false positives in short-read SV detection (Nattestad et al., 2018).

We have also added supplementary figures further characterizing SV lengths and allele frequencies (Figure 1 – S5, Figure 1 – S6), as well as their distribution throughout the genome (Figure 1 – S1). We have expanded our discussion of these findings in the text of the paper (lines 147-153).

2. Related to the above point, the use of the graph genotyping software, Paragraph, is s sensible approach, and I suspect that this paper may provide a template for future analyses merging long-read and short-read SV data. Given this potential contribution, more should be done to discuss the benefits and limitations of this approach. There is brief discussion of "ascertainment bias" based on the 15 individuals with long-read data used from Audano et al., 2019. How does the ancestry of these 15 individuals influence the ability to comprehensively identify SVs in thousands of individuals with different ancestries? A few more details in the main text regarding the sensitivity and specificity of Paragraph would also help to aid result interpretation (rather than just "recent benchmarking…support the accuracy of Paragraph"). Does the Paragraph approach better identify longer SVs, deletions, duplications, inversions, SVs in more repetitive regions of the genome?

Thank you for these questions. Regarding ascertainment bias, the diversity of the long-read discovery set we used (sample ancestries: 3 African, 2 American, 3 East Asian, 2 European, 2 South Asian, 2 hydatidiform moles [likely European]) allows us to discover approximately equivalent numbers of SVs across the five continental superpopulations of 1KGP. We appreciate the importance of including this information in the paper and now discuss the ancestries of the long-read sequenced individuals in lines 77-80.

Thank you also for the questions about the accuracy of Paragraph, which are again closely related to questions from Reviewer 1. We now provide additional results supporting the accuracy of graph genotyping on lines 100-107, including important details of previous benchmarking studies. Briefly, we find that among graph-based and non-graph based genotyping tools, Paragraph consistently attains the best balance of precision and recall for both insertions and deletions. Paragraph averages precision and recall rates of 0.72 and 0.70, respectively, across all SV types and genomic regions, and an average precision and recall of 0.86 and 0.79 when repeat regions are excluded. These metrics support its use for genotyping SVs even in repetitive regions of the genome that are typically difficult to map with short reads (Chen et al., 2019; Hickey et al., 2020).

3. The authors identify 1121 significant associations between SVs and the expression of nearby genes in LCLs. This analysis is reasonable and clear, but the implications are not clear. Prior to this section, the authors discuss LD with previously investigated SNPs/indels, yet they do not report this information for the eQTL 1121 SVs. How many of these 1121 are in high LD with common SNPs/indels? And how many are not linked to previously investigated variation? Even if most are linked, this is still potentially interesting, but it would be helpful to know if they are identified as eQTL in other datasets (GTEx?). Do these potentially SV mediated eQTL have greater effect sizes than eQTL caused by SNVs? Is there a way to fine map one or two (potentially one related to the subsequently immunoglobulin story or the ones highlighted in Figure 2C) to show that the SV is likely driving the association, rather than other linked variants?

In response to these questions, we have added more biological context to our eQTL analysis by intersecting SVs with annotated regulatory elements (lines 215-220). We observed a strong enrichment of significant gene expression associations for SVs that intersect annotated enhancer elements, as well as a strong depletion of expression associations for SVs that intersect “quiescent” sequences that are devoid of important epigenetic marks (Figure 2D). The intuitive nature of these results supports the robustness of our analysis, while also shedding light on functional mechanisms through which SVs mediate impacts on phenotypes and fitness.

We also investigated direct dosage effects of exon-spanning SVs, showing that the magnitude and direction of these effects are broadly consistent with expectations (lines 220-225). We also performed a fine-mapping analysis of SV and SNP eQTLs (lines 228-244). While noting important technical caveats, both of these analyses clarify the causal link between a subset of SVs and the observed gene expression changes.

Finally, we checked whether any frequency-differentiated SVs identified in our study were also identified as eQTLs, and found seven such SVs (Supplementary Table 2). One major limitation of this analysis is the lack of diversity in the Geuvadis dataset, which restricts us to SVs that are common in European or Yoruban populations. Consequently, we were unable to directly test gene expression associations for SVs in the IGH locus in East Asian samples, because this ancestry group is not represented in Geuvadis.

4. The authors use Ohana to test for local adaptation. This method seems appropriate for the challenges of the SV data; the results are interesting; and I appreciate their new individual examples. However, these results are largely descriptive. Testing larger hypotheses about whether SVs are targets of selection more often than SNVs or if these patterns differ between populations or parts of the genome would substantially increase the impact of these results. Figure 4-S1 may be relevant to this kind of comparison; however, it only shows a few loci. While differences in power and ascertainment may make direct comparison challenging, I encourage the authors to think about how to move beyond a list of examples to more general conclusions.

Thank you for this suggestion. We explored the idea of “positive selection fine-mapping” during our initial investigation of the Ohana results, including attempting to adapt CAVIAR (Hormozdiari et al., 2014) toward this goal. However, we were ultimately unsatisfied with the reliability and interpretability of this analysis, which extends beyond the intended use of CAVIAR (fine-mapping of genotype-phenotype associations) and was also sensitive to even low rates of genotyping error, which remains higher for SVs than SNPs. We believe that such selection fine-mapping and enrichment analyses are an interesting area for methodological development that lies outside the scope of our current study, but we agree with the broader goal of moving beyond lists of examples to more general conclusions.

In this spirit, we have identified 133 of the 220 outlier SVs where the SV is among the top 0.1% of variants within a 1 Mb window (lines 290-293). Several other aspects of our revision also extend beyond specific loci (e.g., quantitative comparisons of SV catalogs, eQTL functional enrichments) and should help address this point.

Additionally, the y-axis range on Figure 4 across the different ancestries are quite different. It is hard to interpret to the significance and magnitude of the LRS. Can the multiple testing control be integrated here?

Stratifying the analysis by ancestry component is important, as the test has different power for detecting allele frequency changes on different branches of the tree. A combined comparison of LRS would cause certain ancestry components to dominate the results. See (Cheng et al., 2021) for a description of the method and a demonstration on stratified ancestry groups, which we followed in our work.

Prior to our initial submission, we very carefully considered multiple testing correction and the interpretation of LRS p-values, which could naively be adjusted to control the family-wise error rate or false discovery rate using standard methods. Even using very stringent thresholds, we found this approach to identify implausible numbers of selected loci. However, we note that the very large dataset (thousands of samples) means that even very small improvements in the fit of the more complex model cause it to be favored over the reduced model. In this context, the p-value is not particularly meaningful, except for ranking of results for a given ancestry component.

Drawing a dividing line between positively selected and neutrally evolving loci is a general challenge in the field, which we believe is important but beyond the scope of our work. While evolutionary simulation offers one useful approach, it requires us to specify the parameters of the demographic model including changes in population sizes and migration, which remain uncertain even for individual populations, let alone for a global sample.

Given these challenges, we ultimately settled on an outlier-based approach, which we still find valuable given the extreme signatures observed at loci such as IGH, especially given its known role in adaptive immunity.

Nevertheless, the reviewer’s point is very well taken, and we have bolstered our selection results at IGH by performing 100,000 neutral simulations under the simplified demographic model of East Asia described by (Jouganous et al., 2017; Wang et al., 2018) (lines 494-496; Figure 5 – S1), demonstrating that the observed magnitudes of allele frequency differentiation drastically exceed those expected under genetic drift alone.

5. Finally, the authors explore one example, the immunoglobulin heavy chain locus, in detail. The example and analysis are compelling; however, it is complicated and sometimes hard to follow given the multiple SVs in this region. Readers would benefit from a cartoon schematic of this locus, perhaps expanding on the supplemental figure. I would also appreciate more discussion of why this region might have been filtered out in Browning et al., 2018.

We thank the reviewer for pointing out sources of confusion in our discussion of the IGH locus. We have clarified the structure of the locus by adding a schematic in Figure 4A.

We also further investigated the filtering of this region by Browning et al., Their filtering involved a mask for the Altai Neanderthal sequence, generated by (Prüfer et al., 2017), that removes sites that have low coverage or mapping quality, or are located within tandem repeats or indels (Browning et al., 2018). We now add a supplementary figure showing the masking of the broader IGH region that we highlight (Figure 4 – S4), where 19.3% of the sequence remains after masking (in comparison with 62.6% in the remainder of the q-arm of chromosome 14). We also describe the Browning et al., filtering in more detail in the text (lines 415-417).

The Browning et al., study highlighted a subregion of the IGH locus that is downstream of the region we focus on. The haplotype they discuss reaches high frequencies in European and Asian populations, and is distinct from the haplotype we investigate, which reaches high frequency only in southeast Asia. This is displayed in Figure 4 – S3, where the Browning et al., haplotype is represented in the EUR, EAS, and AMR populations, while the one we investigate is only in the EAS populations (with the exception of JPT). The precise reason why Browning et al., focused on this Eurasian haplotype, rather than the one at high frequency in southeast Asia alone, is unclear. However, we hypothesize that this may be due to differences in how our studies chose populations to search for frequency differences between, or the exclusion of variants on the southeast Asian haplotype by the Altai Neanderthal mask (for example, variants within 10 bp of the insertion SV we highlight or its most frequency differentiated tag SNP are removed by masking). While we believe that an in-depth discussion of the multiple haplotypes at the IGH locus would overly complicate the paper, we provide a more detailed description of the distinction between our result and the Browning et al., paper in lines 407-417.

Reviewer #3 (Recommendations for the authors):The abstract and introduction focus on the size of SVs and the relative "ease" of analysis of SNVs and indels, however, the highlighted variants are indels by the 1000 genomes definition. They also seem to have been discovered in other short-read datasets, (e.g. gnomAD).

Thank you for raising this point, which we agree is vital to address in the paper. While the highlighted SV is present in the gnomAD database and sets of small indels called by 1000 Genomes and HGDP (https://gnomad.broadinstitute.org/variant/14-105622991-G-GGGGGACAGCGTCAGGGACAGGTGGGGACAGCCT?dataset=gnomad_r3), it exhibits a highly skewed allele balance (centered on 0.2, in contrast to the expectation of 0.5). The insertion also has low genotyping rates (31% for CDX in 1000 Genomes, compared with 85% in our study) that lead to lower estimates of its allele frequency in East Asian populations (AF = 0.73 in CDX). These caveats to the SV’s discovery in other datasets underscore the technical challenges of genotyping at this locus with traditional short-read approaches. We have incorporated this information into the text of the paper in lines 374-381.

Paragraph has been previously shown to work well for genotyping SVs, however it is not clear what the accuracy in the low-coverage 1000 genomes samples is. After filtering, while 92,286 variants remain, 25,201 (27%) of these are absent from any of the 1000 genomes samples. It is unclear how the authors distinguish "true negatives" from "false negatives."

Our graph genotyping analysis was based on the more recent high coverage (30x) sequencing of 1000 Genomes samples by the New York Genome Center (Byrska-Bishop et al., n.d.) and is therefore similar to the data on which Paragraph was benchmarked. The description of the 25,201 SVs that are absent from any 1000 Genomes sample is simply a description of the genotyping results (the “0” bin of the allele frequency spectrum) and does not attempt to separate false negatives from true negatives. This large number of “0-frequency” variants is expected, however, due to the abundance of rare variation in human populations and the ascertainment scheme of identifying variants in a distinct sample from that in which they are genotyped. We thank the reviewer for raising this point of confusion and have rephrased the text to clarify (lines 128-131).

In support of Paragraph’s genotyping accuracy, we (1) direct readers to the Paragraph and subsequent benchmarking papers (Hickey et al., 2020), where the method was thoroughly evaluated and shown to be more accurate and faster than other SV genotyping methods (lines 100-107); and (2) demonstrate strong concordance in allele frequencies of graph genotyped variants and frequencies of the same variants detected via long-read sequencing (Figure 1 – S4).

I don't think the LD section adds anything new. This is well reported.And LD cutoff of 0.5 R^2^ is used for introgressed haplotypes versus 0.8^2^ for tag haplotypes in the original motivation. Suggest picking one to be consistent throughout.

While we appreciate that this is not the first study to measure LD between SVs and SNPs, we still think it is important to perform this analysis on this specific dataset, both as a general description of the data and to motivate the goal of identifying novel phenotype associations and positive selection targets that were not tagged by known SNPs. However, we recognize that we could better inform readers of the previous context for these LD results, and have discussed other papers with a similar analysis in lines 177-179.

We thank the reviewer for noting the differences in LD thresholds used in different analyses. We have revised the text (lines 174-177) to also include the lower LD threshold (r^2^ = 0.5) in the tag SNP analysis. Use of this threshold in the introgression analysis serves to capture a broad set of candidate introgressed SVs that can then be verified through further analysis. For the tag SNP analysis, we additionally report results for the more stringent threshold (r^2^ = 0.8).

References

1000 Genomes Project Consortium, Auton A, Brooks LD, Durbin RM, Garrison EP, Kang HM, Korbel JO, Marchini JL, McCarthy S, McVean GA, Abecasis GR. 2015. A global reference for human genetic variation. *Nature* 526:68–74.

Almarri MA, Bergström A, Prado-Martinez J, Yang F, Fu B, Dunham AS, Chen Y, Hurles ME, Tyler-Smith C, Xue Y. 2020. Population Structure, Stratification, and Introgression of Human Structural Variation. *Cell* 182:189–199.e15.

Audano PA, Sulovari A, Graves-Lindsay TA, Cantsilieris S, Sorensen M, Welch AE, Dougherty ML, Nelson BJ, Shah A, Dutcher SK, Warren WC, Magrini V, McGrath SD, Li YI, Wilson RK, Eichler EE. 2019. Characterizing the Major Structural Variant Alleles of the Human Genome. *Cell* 176:663–675.e19.

Browning SR, Browning BL, Zhou Y, Tucci S, Akey JM. 2018. Analysis of Human Sequence Data Reveals Two Pulses of Archaic Denisovan Admixture. *Cell* 173:53–61.e9.

Byrska-Bishop M, Evani US, Zhao X, Basile AO, Abel HJ, Regier AA, Corvelo A, Clarke WE, Musunuri R, Nagulapalli K, Fairley S, Runnels A, Winterkorn L, Lowy-Gallego E, Flicek P, Germer S, Brand H, Hall IM, Talkowski ME, Narzisi G, Zody MC, The Human Genome Structural Variation Consortium. n.d. High coverage whole genome sequencing of the expanded 1000 Genomes Project cohort including 602 trios. doi:10.1101/2021.02.06.430068

Cheng JY, Racimo F, Nielsen R. n.d. Ohana: detecting selection in multiple populations by modelling ancestral admixture components. doi:10.1101/546408

Chen L, Wolf AB, Fu W, Li L, Akey JM. 2020. Identifying and Interpreting Apparent Neanderthal Ancestry in African Individuals. *Cell* 180:677–687.e16.

Chen S, Krusche P, Dolzhenko E, Sherman RM, Petrovski R, Schlesinger F, Kirsche M, Bentley DR, Schatz MC, Sedlazeck FJ, Eberle MA. 2019. Paragraph: a graph-based structural variant genotyper for short-read sequence data. *Genome Biol* 20:291.

Chin C-S, Wagner J, Zeng Q, Garrison E, Garg S, Fungtammasan A, Rautiainen M, Aganezov S, Kirsche M, Zarate S, Schatz MC, Xiao C, Rowell WJ, Markello C, Farek J, Sedlazeck FJ, Bansal V, Yoo B, Miller N, Zhou X, Carroll A, Barrio AM, Salit M, Marschall T, Dilthey AT, Zook JM. 2020. A diploid assembly-based benchmark for variants in the major histocompatibility complex. *Nat Commun* 11:4794.

Hickey G, Heller D, Monlong J, Sibbesen JA, Sirén J, Eizenga J, Dawson ET, Garrison E, Novak AM, Paten B. 2020. Genotyping structural variants in pangenome graphs using the vg toolkit. *Genome Biol* 21:35.

Hormozdiari F, Kostem E, Kang EY, Pasaniuc B, Eskin E. 2014. Identifying causal variants at loci with multiple signals of association. *Genetics* 198:497–508.

Ilardo MA, Moltke I, Korneliussen TS, Cheng J, Stern AJ, Racimo F, de Barros Damgaard P, Sikora M, Seguin-Orlando A, Rasmussen S, van den Munckhof ICL, Ter Horst R, Joosten LAB, Netea MG, Salingkat S, Nielsen R, Willerslev E. 2018. Physiological and Genetic Adaptations to Diving in Sea Nomads. *Cell* 173:569–580.e15.

Jouganous J, Long W, Ragsdale AP, Gravel S. 2017. Inferring the Joint Demographic History of Multiple Populations: Beyond the Diffusion Approximation. *Genetics* 206:1549–1567.

Nattestad M, Goodwin S, Ng K, Baslan T, Sedlazeck FJ, Rescheneder P, Garvin T, Fang H, Gurtowski J, Hutton E, Tseng E, Chin C-S, Beck T, Sundaravadanam Y, Kramer M, Antoniou E, McPherson JD, Hicks J, McCombie WR, Schatz MC. 2018. Complex rearrangements and oncogene amplifications revealed by long-read DNA and RNA sequencing of a breast cancer cell line. *Genome Res* 28:1126–1135.

Prüfer K, de Filippo C, Grote S, Mafessoni F, Korlević P, Hajdinjak M, Vernot B, Skov L, Hsieh P, Peyrégne S, Reher D, Hopfe C, Nagel S, Maricic T, Fu Q, Theunert C, Rogers R, Skoglund P, Chintalapati M, Dannemann M, Nelson BJ, Key FM, Rudan P, Kućan Ž, Gušić I, Golovanova LV, Doronichev VB, Patterson N, Reich D, Eichler EE, Slatkin M, Schierup MH, Andrés AM, Kelso J, Meyer M, Pääbo S. 2017. A high-coverage Neandertal genome from Vindija Cave in Croatia. *Science* 358:655–658.

Schaefer NK, Shapiro B, Green RE. 2021. An ancestral recombination graph of human, Neanderthal, and Denisovan genomes. *Sci Adv*
**7**. doi:10.1126/sciadv.abc0776

Sudmant PH, Rausch T, Gardner EJ, Handsaker RE, Abyzov A, Huddleston J, Zhang Y, Ye K, Jun G, Fritz MH-Y, Konkel MK, Malhotra A, Stütz AM, Shi X, Casale FP, Chen J, Hormozdiari F, Dayama G, Chen K, Malig M, Chaisson MJP, Walter K, Meiers S, Kashin S, Garrison E, Auton A, Lam HYK, Mu XJ, Alkan C, Antaki D, Bae T, Cerveira E, Chines P, Chong Z, Clarke L, Dal E, Ding L, Emery S, Fan X, Gujral M, Kahveci F, Kidd JM, Kong Y, Lameijer E-W, McCarthy S, Flicek P, Gibbs RA, Marth G, Mason CE, Menelaou A, Muzny DM, Nelson BJ, Noor A, Parrish NF, Pendleton M, Quitadamo A, Raeder B, Schadt EE, Romanovitch M, Schlattl A, Sebra R, Shabalin AA, Untergasser A, Walker JA, Wang M, Yu F, Zhang C, Zhang J, Zheng-Bradley X, Zhou W, Zichner T, Sebat J, Batzer MA, McCarroll SA, 1000 Genomes Project Consortium, Mills RE, Gerstein MB, Bashir A, Stegle O, Devine SE, Lee C, Eichler EE, Korbel JO. 2015. An integrated map of structural variation in 2,504 human genomes. *Nature* 526:75–81.

VanLiere JM, Rosenberg NA. 2008. Mathematical properties of the r2 measure of linkage disequilibrium. *Theor Popul Biol* 74:130–137.

Wang Y, Lu D, Chung Y-J, Xu S. 2018. Genetic structure, divergence and admixture of Han Chinese, Japanese and Korean populations. *Hereditas* 155:19.

Yair S, Lee KM, Coop G. 2021. The timing of human adaptation from Neanderthal introgression. *Genetics* 218. doi:10.1093/genetics/iyab052

Zhang J-Y, Roberts H, Flores DSC, Cutler AJ, Brown AC, Whalley JP, Mielczarek O, Buck D, Lockstone H, Xella B, Oliver K, Corton C, Betteridge E, Bashford-Rogers R, Knight JC, Todd JA, Band G. 2021. Using de novo assembly to identify structural variation of eight complex immune system gene regions. *PLoS Comput Biol* 17:e1009254.

Zhao X, Collins RL, Lee W-P, Weber AM, Jun Y, Zhu Q, Weisburd B, Huang Y, Audano PA, Wang H, Walker M, Lowther C, Fu J, Human Genome Structural Variation Consortium, Gerstein MB, Devine SE, Marschall T, Korbel JO, Eichler EE, Chaisson MJP, Lee C, Mills RE, Brand H, Talkowski ME. 2021. Expectations and blind spots for structural variation detection from long-read assemblies and short-read genome sequencing technologies. *Am J Hum Genet* 108:919–928.